# Fill-with-Anything: High-Resolution and Prompt-Faithful Text-Guided Image Inpainting

## Abstract

Building on the achievements of text-to-image diffusion models, recent advancements in text-guided image inpainting have yielded remarkably realistic and visually compelling outcomes. Nevertheless, current text-to-image inpainting models leave substantial room for enhancement, particularly in addressing the often inadequate alignment of user prompts with the inpainted region, and in extending applicability to high-resolution images. To this end, this paper introduces an entirely **training-free** approach that **faithfully adheres to prompts** and seamlessly **scale to high-resolution** image inpainting. To achieve this, we first present the Prompt-Aware Introverted Attention (PAIntA) layer, which enriches self-attention modules by incorporating prompt information derived from cross-attention scores, alleviating the visual context dominance in inpainting caused by all-to-all attention. Furthermore, we introduce the Reweighting Attention Score Guidance (RASG) mechanism, which directs cross-attention scores towards improved textual alignment while preserving the generation domain. In addition, to address inpainting at larger scales, we introduce a specialized super-resolution technique tailored for inpainting, enabling the completion of missing regions in images of up to 2K resolution. Experimental results demonstrate that our proposed method surpasses existing state-of-the-art approaches in both qualitative and quantitative measures, achieving a substantial generation accuracy improvement of **61.4%** compared to **51.9%**. Our codes will be open-sourced.

## 1 Introduction

The recent wave of diffusion models Ho et al. (2020); Song et al. (2021) has taken the world by storm, becoming an increasingly integral part of our everyday lives. After the unprecedented success of text-to-image models Rombach et al. (2022); Ramesh et al. (2022); Saharia et al. (2022); Wu et al. (2022) diffusion-based image manipulations such as prompt-conditioned editing Hertz et al. (2022); Brooks et al. (2023), controllable generation Zhang & Agrawala (2023); Mou et al. (2023), personalized and specialized image synthesis Ruiz et al. (2023); Gal et al. (2022); Lu et al. (2023), etc. became hot topics in computer vision leading to a huge amount of applications. Particularly, text-guided image completion or inpainting Wang et al. (2023b); Wu et al. (2022); Avrahami et al. (2022) allows users to generate new content in user-specified regions of given images based on textual prompts (see Fig. 1), leading to use cases like retouching specific areas of an image, replacing or adding objects, and modifying subject attributes such as clothes, colors, or emotion.

Pretrained text-to-image generation models such as Stable Diffusion Rombach et al. (2022), Imagen Saharia et al. (2022), and Dall-E 2 Ramesh et al. (2022) can be adapted for image completion by blending diffused known regions with generated (denoised) unknown regions during the backward diffusion process. Although such approaches Avrahami et al. (2022; 2023) produce well-harmonized and visually plausible completions, they lack global scene understanding especially when denoising in high diffusion timesteps.

To address this, existing methods Rombach et al. (2022); Wang et al. (2023b); Nichol et al. (2021); Xie et al. (2023) modify pretrained text-to-image models to take additional context information and fine-tune specifically for text-guided image completion. GLIDE (Nichol et al. (2021)) and Stable Inpainting (inpainting method fine-tuned on Stable Diffusion) concatenate the mask and the masked

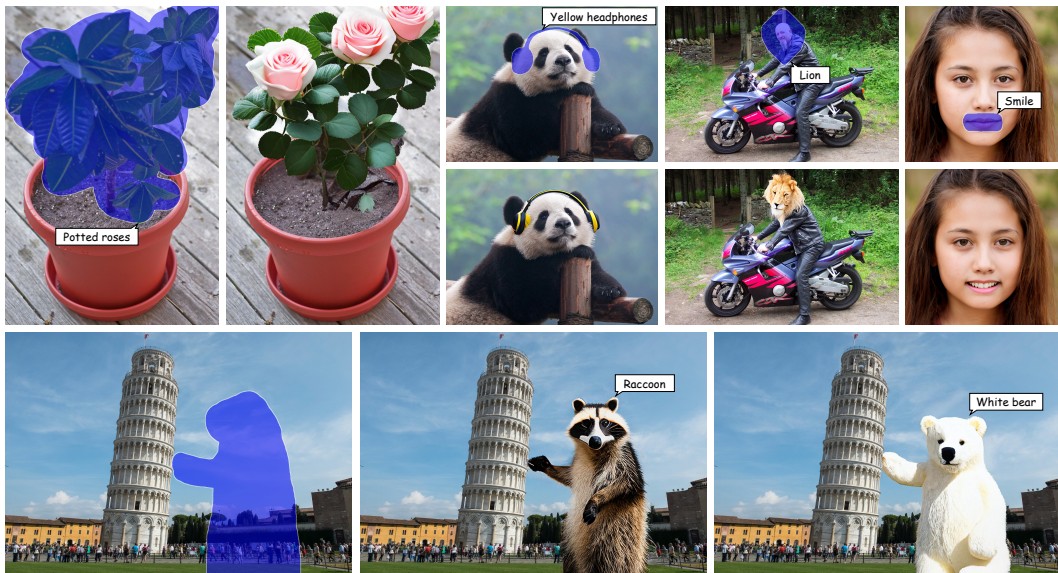

Figure 1: High-resolution (the large side is 2048 in all these examples) text-guided image inpainting results with our approach. The method is able to faithfully fill the masked region according to the prompt even if the combination of the prompt and the known region is highly unlikely.

image as additional channels to the input of the diffusion UNet, initializing the new convolutional weights with zeros. Furthermore, to get better mask alignment, SmartBrush Xie et al. (2023) utilizes instance-aware training with object bounding boxes and segmentation masks.

Despite the mentioned approaches yielding high-quality generation with impressive diversity and realism, we noticed a major drawback of *prompt neglect* expressed by two scenarios: $(i)$ *Background dominance*: when the unknown region is completed with the background ignoring the prompt (e.g. Fig. 4, rows 1, 3), and $(ii)$ *nearby object dominance*: when the known region objects are propagated to the unkown region according to the visual context likelihood rather than the given prompt (e.g. Fig. 4, rows 5, 6).

Perhaps both issues arise because the vanilla diffusion inpainting lacks the capability to accurately interpret the textual prompt or combine it with the contextual information from the known region. To address the mentioned problems we introduce *Prompt-Aware Introverted Attention (PAIntA)* block without any *training or fine-tuning* requirements. PAIntA enhances the self-attention scores according to the given textual condition aiming to decrease the impact of non-prompt-relevant information from the image known region while increasing the contribution of the prompt-aligned known pixels.

To improve the text-alignment of the generation results even further we apply a *post-hoc guidance* mechanism by leveraging the cross-attention scores. However the vanilla post-hoc guidance mechanism used by seminal works such as Dhariwal & Nichol (2021); Epstein et al. (2023), etc. may lead to generation quality degradation due to out-of-distribution shifts caused by the additional gradient term in the backward diffusion equation (see Eq. 4). To this end we propose *Reweighting Attention Score Guidance (RASG)*, a post-hoc mechanism seamlessly integrating the gradient component in the general form of DDIM process. This allows to simultaneously guide the sampling towards more prompt-aligned latents and keep them in their trained domain leading to visually plausible inpainting results.

With the combination of PAIntA and RASG our method gains a significant advantage over the current state-of-the-art approaches by solving the issue of prompt neglect. Moreover PAIntA and RASG both are plug-and-play components so can be added on top of any diffusion base inpainting model to alleviate the mentioned problem. In addition, by leveraging high-resolution diffusion models and time-iterative blending technology we design a simple yet effective pipeline for up to $2048 \times 2048$ resolution inpainting.

To summarize, our main contributions are as follows:

- We introduce the *Prompt-Aware Introverted Attention (PAIntA)* layer to alleviate the prompt neglect issues of background and nearby object dominance in text-guided image inpainting.

- To further improve the text-alignment of generation we present the *Reweighting Attention Score Guidance (RASG)* strategy which enables to prevent out-of-distribution shifts while performing post-hoc guided sampling.

- Our designed pipeline for text-guided image completion is *completely training-free* and demonstrates a significant advantage over current state-of-the-art approaches quantitatively and qualitatively. Moreover, with the additional help of our simple yet effective inpainting-specialized super-resolution framework we make high-resolution (up to $2048{\times}2048$) image completion possible.

## 2 RELATED WORK

### 2.1 IMAGE INPATINING

Image inpainting is the task of filling missing regions of the image in a visually plausible manner. Early deep learning approaches such as Yu et al. (2018); Yi et al. (2020); Navasardyan & Ohanyan (2020) introduce mechanisms to propagate deep features from known regions. Later Zhao et al. (2021); Zheng et al. (2022); Xu et al. (2023); Sargsyan et al. (2023) utilize StyleGAN-v2-like Karras et al. (2020) decoder and discriminative training for better generation of image details. Suvorov et al. (2022) uses Fourier convolutions to perform resolution-robust inpainting.

After diffusion models have been introduced the inpainting task also benefited from them. Particularly text-guided image inpainting approaches emerged. Given a pre-trained text-to-image diffusion model Avrahami et al. (2022; 2023) replace the unmasked region of the latent by the noised version of the known region during sampling. However, as noted by Nichol et al. (2021), this leads to poor generation quality, as the denoising network only sees the noised version of the known region. Therefore they propose fine-tuning pretrained text-to-image models for text-guided image inpainting by conditioning the denoising model on the unmasked region and a generated random mask via concatenation. Wang et al. (2023a) finds that using object masking instead of random masking during fine-tuning improves text-image alignment. Xie et al. (2023) incorporates object-mask prediction into training to better preserve the background given a coarse input mask.

### 2.2 INPAINTING-SPECIFIC ARCHITECTURAL BLOCKS

Early deep learning approaches were designing special layers for better/more efficient inpainting. Particularly, Liu et al. (2018); Yu et al. (2019); Navasardyan & Ohanyan (2020) introduce special convolutional layers dealing with the known region of the image to effectively extract the information useful for visually plausible image completion. Yi et al. (2020) introduces the contextual attention layer reducing the unnecessarily heavy computations of all-to-all self-attention for high-quality inpainting. In this work we propose Prompt-Aware Introverted Attention (PAIntA) layer, specifically designed for text-guided image inpainting. It aims to decrease (increase) the prompt-irrelevant (-relevant) information from the known region for better text aligned inpainting generation.

### 2.3 POST-HOC GUIDANCE IN BACKWARD DIFFUSION PROCESS

Post-hoc guidance methods are backward diffusion sampling techniques which guide the next step latent prediction towards a specific objective function minimization. Such approaches appear to be extremely helpful when generating visual content especially with an additional constraint. Particularly Dhariwal & Nichol (2021) introduced classifier-guidance aiming to generate images of a specific class. Later CLIP-guidance was introduced by Nichol et al. (2021) leveraging CLIP Radford et al. (2021) as an open-vocabulary classification method. LDM Rombach et al. (2022) further extends the concept to guide the diffusion sampling process by any image-to-image translation method, particularly guiding a low-resolution trained model to generate $\times2$ larger images. Chefer et al. (2023) guides image generation by maximizing the maximal cross-attention score relying on multi-iterative optimization process resulting in more text aligned results. Epstein et al. (2023) goes even further by utilizing the cross-attention scores for object position, size, shape, and appearance guidances. All the mentioned post-hoc guidance methods shift the latent generation process by a gradient term (see Eq. 6) sometimes leading to image quality degradations.

To this end we propose Reweighting Attention Score Guidance (RASG) mechanism allowing post-hoc guidance with any objective function **while preserving the diffusion latent domain**. Specifically for inpainting, to alleviate the issue of prompt neglect, we found it beneficial to design our guidance objective function based on the open-vocabulary segmentation properties of cross-attentions.

## 3 METHOD

We first formulate the text-guided image completion problem followed by an introduction to diffusion models, particularly Stable Diffusion (Rombach et al. (2022)) and Stable Inpainting. We then discuss the overview of our method and its components. Afterwards we present our Prompt-Aware Introverted Attention (PAIntA) block and Reweighting Attention Score Guidance (RASG) mechanism in detail. Lastly our inpainting-specific super-resolution technique is introduced.

Let $I \in \mathbb{R}^{H \times W \times 3}$ be an RGB image, $M \in \{0,1\}^{H \times W}$ be a binary mask indicating the region in $I$ one wants to inpaint with a textual prompt $\tau$. The goal of text-guided image inpainting is to output an image $I^c \in \mathbb{R}^{H \times W \times 3}$ such that $I^c$ contains the objects described by the prompt $\tau$ in the region $M$ while outside $M$ it coincides with $I$, i.e. $I^c \odot (1-M) = I \odot (1-M)$.

### 3.1 STABLE DIFFUSION AND STABLE INPAINTING

Stable Diffusion (SD) is a diffusion model that functions within the latent space of an autoencoder $\mathcal{D}(\mathcal{E}(\cdot))$ (VQ-GAN Esser et al. (2021) or VQ-VAE Van Den Oord et al. (2017)) where $\mathcal{E}$ denotes the encoder and $\mathcal{D}$ the corresponding decoder. Specifically, let $I \in \mathbb{R}^{H \times W \times 3}$ be an image and $x_0 = \mathcal{E}(I)$, consider the following forward diffusion process with hyperparameters $\{\beta_t\}_{t=1}^T \subset [0,1]$:

$$q(x_t|x_{t-1}) = \mathcal{N}(x_t; \sqrt{1-\beta_t}x_{t-1}, \beta_t I), \ t = 1, .., T \tag{1}$$

where $q(x_t|x_{t-1})$ is the conditional density of $x_t$ given $x_{t-1}$, and $\{x_t\}_{t=0}^T$ is a Markov chain. Here $T$ is large enough to allow an assumption $x_T \sim \mathcal{N}(\mathbf{0}, \mathbf{1})$. Then SD learns a backward process (below similarly, $\{x_t\}_{t=T}^0$ is a Markov chain)

$$p_\theta(x_{t-1}|x_t) = \mathcal{N}(x_{t-1}; \mu_\theta(x_t, t), \sigma_t \mathbf{1}) \quad \text{for } t = T, \ldots, 1, \tag{2}$$

and hyperparameters $\{\sigma_t\}_{t=1}^T$, allowing the generation of a signal $x_0$ from the standard Gaussian noise $x_T$. Here $\mu_\theta(x_t, t)$ is defined by the predicted noise $\epsilon_\theta^t(x_t)$ modeled as a neural network (see Ho et al. (2020)): $\mu_\theta(x_t, t) = \frac{1}{\sqrt{\beta_t}}\left(x_t - \frac{\beta_t}{\sqrt{1-\alpha_t}}\epsilon_\theta^t(x_t)\right)$. Then $\hat{I} = \mathcal{D}(x_0)$ is returned.

The following claim can be derived from the main DDIM principle, Theorem 1 in Song et al. (2021).

CLAIM **1** *After training the diffusion backward process (Eq. 2) the following $\{\sigma_t\}_{t=1}^T$-parametrized family of DDIM sampling processes can be applied to generate high-quality images:*

$$x_{t-1} = \sqrt{\alpha_{t-1}}\frac{x_t - \sqrt{1-\alpha_t}\epsilon_\theta^t(x_t)}{\sqrt{\alpha_t}} + \sqrt{1-\alpha_{t-1}-\sigma_t^2}\epsilon_\theta^t(x_t) + \sigma_t\epsilon_t, \tag{3}$$

*where $\epsilon_t \sim \mathcal{N}(\mathbf{0}, \mathbf{1})$, $\alpha_t = \prod_{i=1}^t(1-\beta_i)$, and $0 \leq \sigma_t \leq \sqrt{1-\alpha_{t-1}}$ can be arbitrary parameters.*

Usually (e.g. in Stable Diffusion or Stable Inpainting described below) $\sigma_t = 0$ is taken to get a deterministic sampling process:

$$x_{t-1} = \sqrt{\alpha_{t-1}}\left(\frac{x_t - \sqrt{1-\alpha_t}\epsilon_\theta^t(x_t)}{\sqrt{\alpha_t}}\right) + \sqrt{1-\alpha_{t-1}}\epsilon_\theta^t(x_t), \quad t = T, \ldots, 1. \tag{4}$$

For text-to-image synthesis, SD guides the processes with a textual prompt $\tau$. Hence the function $\epsilon_\theta^t(x_t) = \epsilon_\theta^t(x_t, \tau)$, modeled by a UNet-like (Ronneberger et al. (2015)) architecture, is also conditioned on $\tau$ by its cross-attention layers. For simplicity sometimes we skip $\tau$ in writing $\epsilon_\theta^t(x_t, \tau)$.

As mentioned earlier, Stable DIffusion can be modified and fine-tuned for text-guided image inpainting. To do so Rombach et al. (2022) concatenate the features of the masked image $I^M = I \odot (1-M)$ obtained by the encoder $\mathcal{E}$, and the (downscaled) binary mask $M$ to the latents $x_t$ and feed the resulting tensor to the UNet to get the estimated noise $\epsilon_\theta^t([x_t, \mathcal{E}(I^M), down(M)], \tau)$, where $down$ is the downscaling operation to match the shape of the latent $x_t$. Newly added convolutional filters are initialized with zeros while the rest of the UNet from a pretrained checkpoint of Stable Diffusion. Training is done by randomly masking images and optimizing the model to reconstruct them based on image captions from the LAION-5B (Schuhmann et al. (2022)) dataset. The resulting model shows visually plausible image completion and we refer to it as *Stable Inpainting*.

## 3.2 HD-PAINTER: OVERVIEW

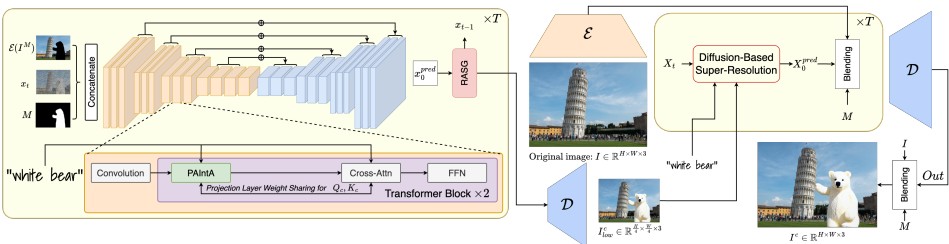

Figure 2: Our method has two stages: image completiton, and inpainting-specialized super-resolution ($\times 4$). For image completion in each diffusion step we denoise the latent $x_t$ by conditioning on the inpainting mask $M$ and the masked downscaled image $I^M = down(I) \odot (1-M) \in \mathbb{R}^{\frac{H}{4} \times \frac{W}{4} \times 3}$ (encoded with the VAE encoder $\mathcal{E}$). To make better alignement with the given prompt our *PAIntA* block is applied instead of self-attention layers. After predicting the denoised $x_0^{pred}$ in each step $t$, we provide it to our *RASG* guidance mechanism to estimate the next latent $x_{t-1}$. For inpainting-specific super resolution we condition the high-resolution latent $X_t$ denoising process by the lower resolution inpainted result $I_{low}^c$, followed by blending $X_0^{pred} \odot M + \mathcal{E}(I) \odot (1-M)$. Finally we get $I^c$ by Poisson blending the decoded output with the original image $I$.

The overview of our method is presented in Fig. 2. The proposed pipeline is composed of two stages: text-guided image inpainting on the resolution $H/4 \times W/4$ is applied followed by the inpainting-specific $\times 4$ super-resolution of the generated content.

To complete the missing region $M$ according to the given prompt $\tau$ we take a pre-trained inpainting diffusion model like Stable Inpainting, replace the self-attention layers by PAIntA layers, and perform a diffusion backward process by applying our RASG mechanism. After getting the final estimated latent $x_0$, it is decoded resulting in an inpainted image $I_{low}^c = \mathcal{D}(x_0) \in \mathbb{R}^{\frac{H}{4} \times \frac{W}{4}}$.

To inpaint the original size image $I \in \mathbb{R}^{H \times W}$ we utilize the super-resolution stable diffusion from Rombach et al. (2022). We apply the diffusion backward process of SD starting from $X_T \sim \mathcal{N}(\mathbf{0}, \mathbf{1})$ and conditioned on the low resolution inpainted image $I_{low}^c$. After each step we blend the denoised $X_0^{pred}$ with the original image's encoding $\mathcal{E}(I)$ in the known region indicated by the mask $(1-M) \in \{0,1\}^{H \times W}$ and derive the next latent $X_{t-1}$ by Eq. 4. After the final step we decode the latent by $\mathcal{D}(X_0)$ and use Poisson blending (Pérez et al. (2023)) with $I$ to avoid edge artifacts.

### 3.3 PROMPT-AWARE INTROVERTED ATTENTION (PAIntA)

Throughout our experiments we noticed that existing approaches, such as Stable Inpainting, tend to ignore the user-provided prompt relying more on the visual context around the inpainting area. In the introdution we categorized this issue into two classes based on user experience: *background dominance* and *nearby object dominance*. Indeed, for example in Fig. 4, rows 1, 3, 4, the existing solutions (besides BLD) fill the region with background, and in rows 5, 6, they prefer to continue the animal and the car instead of generating a boat and flames respectively. We hypothesize that the *visual context dominance* over the prompt is attributed to the *prompt-free, only-spatial* nature of self-attention layers. To support this we visualize the self-attention scores (see Appendix) and observe a high similarity between the inpainted tokens and such known tokens of the image which have low similarity with the prompt (for more details see Appendix). Therefore, to alleviate the issue, we introduce a plug-in replacement for self-attention, Prompt-Aware Introverted Attention (PAIntA, see Fig. 3 (a)) which utilizes the inpainting mask $M$ and cross-attention matrices to control the self-attention output in the unknown region. Below we discuss PAIntA in detail.

Let $X \in \mathbb{R}^{(h \times w) \times d}$ be the input tensor of PAIntA. Similar to self-attention, PAIntA first applies projection layers to get the queries, keys, and values we denote by $Q_s, K_s, V_s \in \mathbb{R}^{(h \times w) \times d}$ respectively, and the similarity matrix $A_{self} = \frac{Q_s K_s^T}{\sqrt{d}} \in \mathbb{R}^{hw \times hw}$. Then we mitigate the too strong influence of the known region over the unknown by adjusting the attention scores of known pixels contributing to

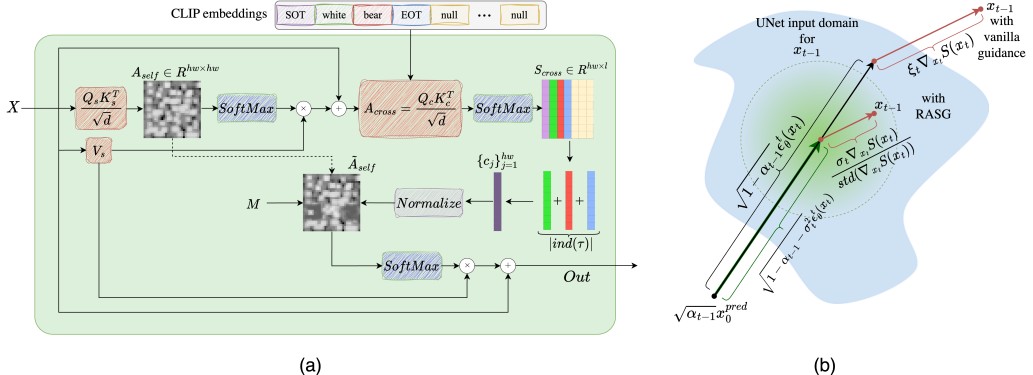

(a)                                                               (b)

Figure 3: (a) PAIntA block takes an input tensor $X \in \mathbb{R}^{h \times w \times 3}$ and the CLIP embeddings of $\tau$. After computing the self- and cross-attention scores $A_{self}$ and $A_{cross}$, we update the former (Eq. 5) by scaling with the normalized values $\{c_j\}_{j=1}^{hw}$ obtained from $S_{cross} = SoftMax(A_{cross})$. Finally the the updated attention scores $\tilde{A}_{self}$ are used for the convex combination of the values $V_s$ to get the residual of PAIntA's output. (b) RASG mechanism takes the predicted scaled denoised latent $\sqrt{\alpha_{t-1}}x_0^{pred} = \frac{\sqrt{\alpha_{t-1}}}{\sqrt{\alpha_t}}\left(x_t - \sqrt{1-\alpha_t}\epsilon_\theta(x_t)\right)$ and guides the $x_{t-1}$ estimation process towards minimization of $S(x_t)$ defined by Eq. 9. Gradient reweighting makes the gradient term close to being sampled from $\mathcal{N}(\mathbf{0}, \mathbf{1})$ (green area) by so ensuring the domain preservation (blue area).

the inpainted region. Specifically, leveraging the prompt $\tau$, PAIntA defines a new similarity matrix:

$$\tilde{A}_{self} \in \mathbb{R}^{hw \times hw}, \quad (\tilde{A}_{self})_{ij} = \begin{cases} c_j \cdot (A_{self})_{ij} & M_i = 1 \text{ and } M_j = 0, \\ (A_{self})_{ij} & \text{otherwise,} \end{cases} \quad (5)$$

where $c_j$ shows the alignment of the $j^{th}$ feature token (pixel) with the given textual prompt $\tau$.

We define $\{c_j\}_{j=1}^{hw}$ using the cross-attention spatio-textual similarity matrix $S_{cross} = SoftMax(Q_c K_c^T / \sqrt{d})$, where $Q_c \in \mathbb{R}^{(h \times w) \times d}$, $K_c \in \mathbb{R}^{l \times d}$ are query and key tensors of corresponding cross-attention layers, and $l$ is the number of tokens of the prompt $\tau$. Specifically, we consider CLIP text embeddings of the prompt $\tau$ and separate the ones which correspond to the words of $\tau$ and *End of Text* (EOT) token (in essence we just disregard the SOT token and the null-token embeddings), and denote the set of chosen indices by $ind(\tau) \subset \{1, 2, \ldots, l\}$. We include EOT since (in contrast with SOT) it contains information about the prompt $\tau$ according to the architecture of CLIP text encoder. For each $j^{th}$ pixel we define its similarity with the prompt $\tau$ by summing up it's similarity scores with the embeddings indexed from $ind(\tau)$, i.e. $c_j = \sum_{k \in ind(\tau)}(S_{cross})_{jk}$. Also, we found beneficial to normalize the scores $c_j = clip\left(\frac{c_j - median(c_k; k=1,\ldots,hw)}{max(c_k; k=1,\ldots,hw)}, 0, 1\right)$, where $clip$ is the clipping operation between $[0, 1]$.

Note that in vanilla SD cross-attention layers come after self-attention layers, hence in PAIntA to get query and key tensors $Q_c, K_c$ we borrow the projection layer weights from the next cross-attention module (see Fig. 2). Finally we get the output of the PAIntA layer with the residual connection with the input: $Out = X + SoftMax(\tilde{A}_{self}) \cdot V_s$.

### 3.4 REWEIGHTING ATTENTION SCORE GUIDANCE (RASG)

To further enhance the generation alignment with the prompt $\tau$ we adopt a post-hoc sampling guidance mechanism Dhariwal & Nichol (2021) with an objective function $S(x)$ leveraging the open-vocabulary segmentation properties of cross-attention layers (we will define it later). More precisely[1] at each step the following update rule is used after predicting the noise $\epsilon_\theta^t(x_t)$: $\hat{\epsilon}_\theta^t(x_t) \leftarrow \epsilon_\theta^t(x_t) + \sqrt{1-\alpha_t} \cdot s\nabla_{x_t}S(x_t)$, where $s$ is a hyperparameter controlling the amount of the guidance. However, as also noted by Chefer et al. (2023), vanilla post-hoc guidance may shift the domain of diffusion latents $x_{t-1}$ resulting in image quality degradations. Indeed, according to the (determinis-

---

[1]for the writing simplicity we skip the conditions $M, I^M, \tau : \epsilon_\theta^t(x_t) = \epsilon_\theta^t([x_t, \mathcal{E}(I^M), down(M)], \tau)$.

tic) DDIM process (Eq. 4) after substituting $\epsilon_\theta^t(x_t)$ with $\hat{\epsilon}_\theta^t(x_t)$ we get

$$x_{t-1} = \sqrt{\alpha_{t-1}}\frac{x_t - \sqrt{1-\alpha_t}\epsilon_\theta^t(x_t)}{\sqrt{\alpha_t}} + \sqrt{1-\alpha_{t-1}}\epsilon_\theta^t(x_t) + \xi_t\nabla_{x_t}S(x_t),$$

$$\xi_t = \sqrt{1-\alpha_t} \cdot s\left(\frac{\sqrt{\alpha_{t-1}}}{\sqrt{\alpha_t}} + \sqrt{1-\alpha_{t-1}}\right),$$
(6)

hence in Eq. 4 the additional term $\xi_t\nabla_{x_t}S(x_t)$ which may shift the original distribution of $x_{t-1}$.

To this end we introduce the *Reweighting Attention Score Guidance (RASG)* strategy which benefits from the general DDIM backward process (Eq. 3) and introduces a gradient reweighting mechanism resulting in latent domain preservation. Specifically, according to Claim 1, $x_{t-1}$ obtained either by Eq. 4 or by Eq. 3 will be in the required domain (see Fig. 3). Hence if in Eq. 3 we replace the stochastic component $\epsilon_t$ by the rescaled version of the gradient $\nabla_{x_t}S(x_t)$ (to make it closer to a sampling from $\mathcal{N}(\mathbf{0},\mathbf{1})$), we will keep $x_{t-1}$ in the required domain and at the same time will guide its sampling towards minimization of $S(x_t)$. Rescaling of the gradient $\nabla_{x_t}S(x_t)$ is done by dividing it on its standard deviation (we do not change the mean to keep the direction of the $S(x_t)$ minimization, for more discussion see Appendix). Thus, sampling with RASG is done by the following formula:

$$x_{t-1} = \sqrt{\alpha_{t-1}}\frac{x_t - \sqrt{1-\alpha_t}\epsilon_\theta(x_t)}{\sqrt{\alpha_t}} + \sqrt{1-\alpha_{t-1}-\sigma_t^2}\epsilon_\theta(x_t) + \sigma_t\frac{\nabla_{x_t}S(x_t)}{\text{std}(\nabla_{x_t}S(x_t))}. \quad (7)$$

Now let us define the objective function $S(x_t)$ (more discussion on its choice can be found in Appendix). First we consider all cross-attention maps $A_{cross}$ with the output resolution of $H/32 \times W/32$: $A_{cross}^1, \ldots, A_{cross}^m \in \mathbb{R}^{(H/32 \cdot W/32) \times l}$, where $m$ is the number of such cross-attention layers, and $l$ is the number of text embeddings. Then for each $k \in ind(\tau) \subset \{1, \ldots, l\}$ we average the attention maps and reshape to $H/32 \times W/32$:

$$\overline{A}_{cross}^k(x_t) = \frac{1}{m}\sum_{i=1}^m A_{cross}^i[:,k] \in \mathbb{R}^{H/32 \times W/32}, \; k \in ind(\tau) \quad (8)$$

Using post-hoc guidance with $S(x_t)$ we aim to maximize the attention scores in the unknown region determined by the binary mask $M \in \{0,1\}^{H \times W}$, hence we take the average binary cross entropy between $\overline{A}^k(x_t)$ and $M$ ($M$ is downscaled with NN interpolation, $\sigma$ is the sigmoid funciton):

$$S(x_t) = -\sum_{k \in ind(\tau)}\sum_{i=1}^{H/32 \cdot W/32}[M_i \log\sigma(\overline{A}_{cross}^k(x_t)_i) + (1-M_i)\log(1-\sigma(\overline{A}_{cross}^k(x_t)_i))]. \quad (9)$$

### 3.5 INPAINTING-SPECIALIZED CONDITIONAL SUPER-RESOLUTION

Here we discuss our method for high-resolution inpainting utilizing a pre-trained diffusion-based super-resolution model. We leverage the fine-grained information from the known region to upscale the inpainted region (see Fig. 2.). Recall that $I \in \mathbb{R}^{H \times W \times 3}$ is the original high-resolution image we want to inpaint, and $\mathcal{E}$ is the encoder of VQ-GAN Esser et al. (2021). We consider $X_0 = \mathcal{E}(I)$ and take a standard Gaussian noise $X_T \in \mathbb{R}^{\frac{H}{4} \times \frac{W}{4} \times 4}$. Then we apply a backward diffusion process (Eq. 4) on $X_T$ by using the upscale-specialized SD model and conditioning it on the low resolution inpainted image $I_{low}^c$. After each diffusion step we blend the estimated denoised latent code $X_0^{pred} = (X_t - \sqrt{1-\alpha_t}\epsilon_\theta^t(X_t))/\sqrt{\alpha_t}$ with $X_0$ by using $M$:

$$X_0^{pred} \leftarrow M \odot X_0^{pred} + (1-M) \odot X_0, \quad (10)$$

and use the new $X_0^{pred}$ to determine the latent $X_{t-1}$ (by Eq. 4). After the last diffusion step $X_0^{pred}$ is decoded and blended (Poisson blending) with the original image $I$.

It's worth noting that our blending approach is inspired by seminal works Sohl-Dickstein et al. (2015); Avrahami et al. (2022) blending $X_t$ with the noisy latents of the forward diffusion. In contrast, we blend high-frequencies from $X_0$ with the denoised prediction $X_0^{pred}$ allowing noise-free image details propagate from the known region to the missing one during all diffusion steps.

| Model Name | CLIP $\uparrow$ | Acc $\uparrow$ | PickScore (Ours vs Baselines) $\downarrow$ |
|---|---|---|---|
| GLIDE | 24.92 | 44.0 % | 62.7 % |
| NUWA-LIP | 24.07 | 31.8 % | 67.5 % |
| BLD | 24.81 | 49.8 % | 56.3 % |
| Stable Inpainting | 24.86 | 51.9 % | 54.3 % |
| Ours | **26.34** | **61.4** % | **50.0** % |

Table 1: Quantitative comparison

| Model Name | CLIP $\uparrow$ | Acc $\uparrow$ |
|---|---|---|
| base (Stable Inpainting) | 24.86 | 51.9 % |
| only PAIntA | 25.24 | 52.2 % |
| only RASG | 25.85 | **62.0** % |
| RASG & PAIntA (Ours) | **26.34** | 61.4 % |

Table 2: Ablation study for PAIntA and RASG.

## 4 EXPERIMENTS

### 4.1 IMPLEMENTATION DETAILS

Our code is based on the Stable Diffusion 2.0 public GitHub repository from `https://github.com/Stability-AI/stablediffusion` which also includes the Stable Inpainting 2.0 and Stable Super-Resolution 2.0 pre-trained models we use as image completion and inpainting-specialized super-resolution baselines respectively. PAIntA is used to replace the self attention layers on the $H/32 \times W/32$ and $H/16 \times W/16$ resolutions. For RASG we select only cross-attention similarity matrices of the $H/32 \times W/32$ resolution as we noticed no further improvements when taking also finer resolutions (the reason is that the segmentation by cross-attention layers is coarse anyway majorly independent on the layer input resolution) while the diffusion process slows down significantly. For hyperparameters $\{\sigma_t\}_{t=1}^T$ we chose $\sigma_t = \eta \sqrt{\frac{(1-\alpha_{t-1})}{(1-\alpha_t)}} \sqrt{1 - \frac{\alpha_t}{\alpha_{t-1}}}$, $\eta = 0.25$.

### 4.2 EXPERIMENTAL SETUP

Here we compare with existing state-of-the-art methods such as GLIDE Nichol et al. (2021), Stable Inpainting Rombach et al. (2022), NUWA-LIP Ni et al. (2023), and Blended Latent Diffusion (BLD) Avrahami et al. (2023). We evaluate the methods on a random sample of 10000 (image, mask, prompt) triplets from the validation set of MSCOCO 2017 Lin et al. (2014), where the prompt is chosen as the label of the selected instance mask. We noticed that when a precise mask of a recognizable shape is given to Stable Inpainting, it tends to ignore the prompt and inpaint based on the shape. To prevent this, we use the convex hulls of the object segmentation masks and compute the metrics accordingly.

We evaluate the CLIP score on a cropped region of the image using the bounding box of the input mask. Since CLIP score can still assign high scores to adversarial examples, we propose to additionally compute the generation class accuracy. For that, we utilize a pre-trained instance detection model for MSCOCO: MMDetection (Chen et al. (2019)). We run MMDetection on the cropped area of the generated image, and, as there might be more than one objects included in the crop, we treat the example as positive if the prompt label is in the detected object list.

Finally, we employ PickScore (Kirstain et al. (2023)) as a combined metric of text-alignment and visual fidelity. Being trained on real user feedback PickScore is able to not only assess the prompt-faithfulness of inpainting methods but also the generation quality, while reflecting the complex requirements of users from the text-guided inpainting task. In our setting we apply PickScore between our vs other methods results and compute the percentage when it gives the advantage to our.

### 4.3 QUANTITATIVE AND QUALITATIVE ANALYSIS

Table 1 shows that our method outperforms the competitors with all three metrics with a large margin. Particularly we improve by more than **1.5** points of CLIP score on top of all competitors, and reached generation accuracy (**Acc**) of **61.4 %** vs **51.9 %** from other state-of-the-art methods. Furthermore, the PickScore comparison shows that our method is better than the other competitors also in terms of generation quality. We also performed a user study which demonstrates our clear advantage over the competitor state-of-the-art methods in terms of both: *prompt alignment* and *overall quality*. Due to space limitations we present the details of our user study in Appendix.

The examples in Fig. 4 demonstrate qualitative comparison between our method and the other state-of-the-art approaches. In many cases the baseline methods either generate a background (Fig. 4, rows 1, 2, 3, 4) or reconstruct the missing regions as continuation of the known region objects disregarding the prompt (Fig. 4, rows 5,6), while our method, thanks to the combination of PAIntA and RASG, successfully generates the target objects. Notice that BLD generates the required object more than other competitors, however the quality of the generation is poor. Similarly, Stable Inpainting is able to generate the object sometimes but the frequency of getting the prompt-alignment is low (see Appendix).

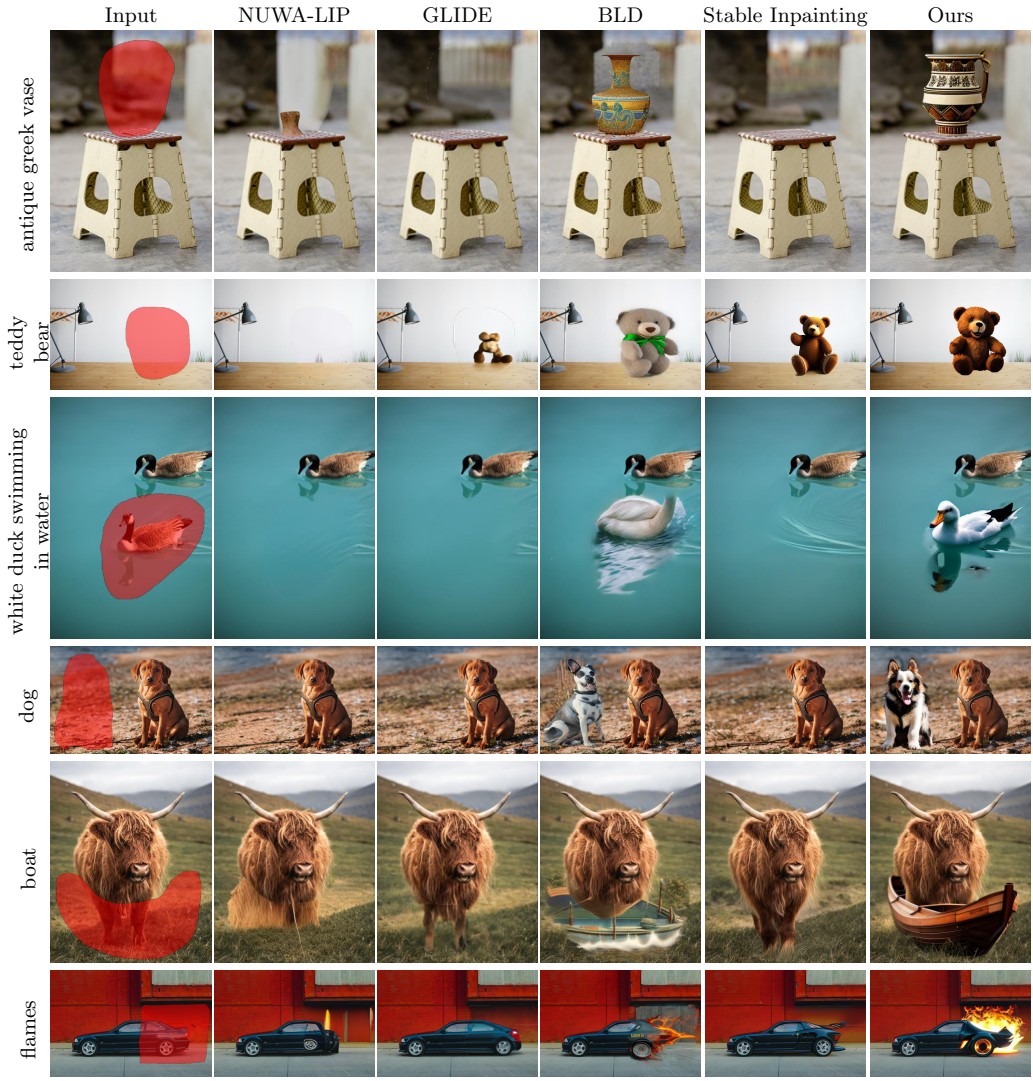

Figure 4: Comparison with recent state-of-the-art text-guided inpainting methods. For more comparison see Appendix.

## 4.4 ABLATION STUDY

In Table 2 we show that PAIntA and RASG separately on their own provide substantial improvements to the model quantitatively. We also provide more discussion on each of them in our supplementary material, including thorough analyses on their impact, demonstrated by visuals.

## 5 CONCLUSION

In this paper, we introduced a training-free approach to text-guided high-resolution image inpainting, addressing the prevalent challenges of prompt neglect: background and nearby object dominance. Our contributions, the Prompt-Aware Introverted Attention (PAIntA) layer and the Reweighting Attention Score Guidance (RASG) mechanism, effectively mitigate the mentioned issues leading our method to surpass the existing state-of-the-art approaches qualitatively and quantitatively. Additionally, our unique inpainting-specific super-resolution technique offers seamless completion in high-resolution images, distinguishing our method from existing solutions.

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
