

Figure 5: Total votes of each method based on our user study for questions a) *Which results best match user prompt?* and b) *Which results have the best overall quality (considering both prompt alignment and image quality)?* The user study shows a clear advantage of our method.

## APPENDIX A    EXTENDED QUALITATIVE COMPARISON

Here in Fig. 13 we show more visual comparison with the other state-of-the-art methods. Fig. 14 includes more comparison on the validation set of MSCOCO 2017 Lin et al. (2014). The results show the advantage of our method over the baselines.

In Fig. 15 we compare our inpainting-specialized super-resolution method with vanilla approaches of Bicubic or Stable Super-Resolution-based upscaling of the inpainting results followed by Poisson blending in the unknown region. We can clearly see that our method, leveraging the known region fine-grained information, can seamlessly fill in with high quality. In Figures 16 and 17 we show more visual comparison between our method and the approach of Stable Super-Resolution.

## APPENDIX B    ROBUSTNESS TO THE RANDOM SEED

Since diffusion-based models generate output images by denoising a randomly sampled starting noise, the resulting outputs are stochastic, and depend on the sampled starting noise. To verify that the insights formulated during the visual comparison with the baselines are consistent, and are not a mere consequence of random chance, we repeated several generations for five randomly sampled starting seeds. In Fig. 19, 20 and 21 we show the results for five different seeds. As can be noticed our method is able to generate the prompt-aligned objects regardless the initial noise, whereas the competitors perform the desired generation not so frequently.

## APPENDIX C    USER STUDY

In addition to the qualitative/quantitative comparisons with the existing state-of-the-art approaches, we also performed a user study. We included 10 people in the study who were asked to assess 5 text-guided inpainting methods: NUWA-LIP Ni et al. (2023), GLIDE Nichol et al. (2021), BLD Avrahami et al. (2023), Stable Inpainting Rombach et al. (2022), and Ours. The participants were provided with 20 *(image, mask, prompt)* triplets and the inpainting results of the 5 methods in the random order, and for each image were asked the following questions:

(a) *Which results best match user prompt?*

(b) *Which results have the best overall quality (considering both prompt alignment and image quality)?*

The participants were allowed to choose none or multiple best methods since some approaches can perform equally good or all of them be equally bad. After collecting the feedback from the users we calculate the total votes for all the methods and for both questions. The results are presented in Fig. 5 demonstrating a clear advantage of our method in both aspects: prompt alignment and overall quality.

## APPENDIX D    DISCUSSION ON PAINTA

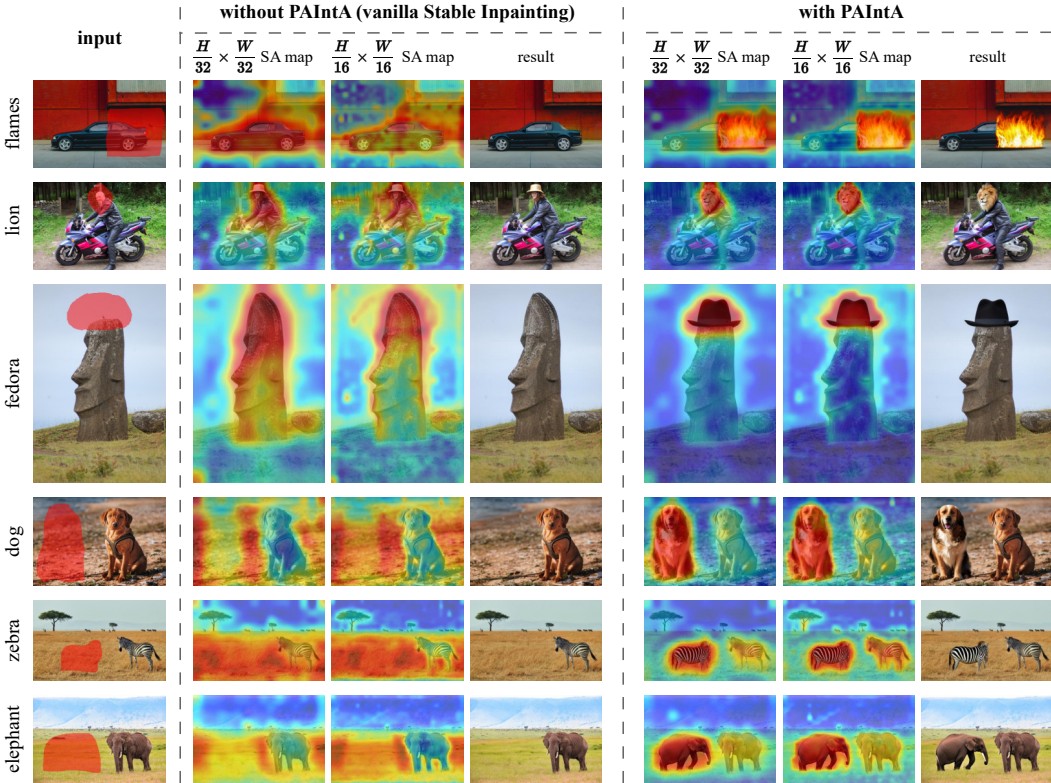

Figure 6: Comparison of self-attention similarity maps averaged across masked pixels for generations without/with PAIntA's scaling of the original self-attention scores.

In this section we discuss the effectiveness of the proposed PAIntA module as a plug-in replacement for self-attention (SA) layers. To that end, first we visualize SA similarity maps averaged across masked locations from resolutions $H/16 \times W/16$ and $H/32 \times W/32$ where PAIntA is applied (see Fig. 6). Then, we see that PAIntA successfully scales down the similarities of masked locations with prompt-unrelated locations from the known region, and, as a result, a prompt-specified object is generated inside the mask.

For a given resolution ($H/16 \times W/16$ or $H/32 \times W/32$), in order to visualize the average SA similarity map across masked pixels, first we resize the input mask to match the dimensions of the corresponding resolution (we use nearest interpolation in resize operation). Then, for each SA layer in the given resolution, we form a 2D similarity map by reshaping and averaging the similarity matrix rows corresponding to the masked region. Further, we average obtained 2D similarity maps across all SA layers (of the given resolution) and diffusion timesteps. More specifically, if $A_{self}^1, \ldots, A_{self}^L \in \mathbb{R}^{hw \times hw}$ ($h \times w$ is either $H/16 \times W/16$ or $H/32 \times W/32$) are the self-attention matrices of Stable Inpainting layers of the given resolution, and, respectively, are being updated by PAIntA to the matrices $\tilde{A}^i{}_{self}$ (see Eq. 5), then we consider the following similarity maps:

$$A = \frac{1}{|M| \cdot L} \sum_{i, M_i = 1} \sum_{l=1}^{L} (A_{self}^l)_i \in \mathbb{R}^{hw},$$

$$\tilde{A} = \frac{1}{|M| \cdot L} \sum_{i, M_i = 1} \sum_{l=1}^{L} (\tilde{A}_{self}^l)_i \in \mathbb{R}^{hw},$$

and reshape them to 2D matrices of size $h \times w$. So, $A_{ij}$ and $\tilde{A}_{ij}$ show the average amount in which masked pixels attend to to other locations in the cases of the vanilla self-attention and PAIntA

respectively. Finally, in order to visualize the similarity maps, we use bicubic resize operation to match it with the image dimensions and plot the similarity heatmap using JET colormap from OpenCV (Itseez, 2015).

Next, we compare the generation results and corresponding similarity maps obtained from above procedure when PAIntA's SA scaling is (the case of $\tilde{A}$) or is not (the case of $A$) used. Because PAIntA's scaling is only applied on $H/32 \times W/32$ and $H/16 \times W/16$ resolutions, we are interested in those similarity maps. Rows 1-3 in Fig. 6 demonstrate visualizations on *nearby object dominance* issue (when known objects are continued to the inpainted region while ignoring the prompt) of the vanilla diffusion inpainting, while rows 4-6 demonstrate those of with *background dominance* issue (when nothing is generated, just the background is coherently filled in).

For example, on row 1, Fig. 6 in case of *Stable Inpainting without PAIntA* generation, the average similarity of the masked region is dominated by the known regions of the car on both 16 and 32 resolutions. Whereas, as a result of PAIntA scaling application, the average similarity of the masked region with the car is effectively reduced, and the masked region is generated in accordance to the input prompt.

Row 4, Fig. 6 demonstrates an example where known region contains a dog, which is aligned with the input prompt. In this case, visualization shows that PAIntA successfully reduces the similarity of the masked region with the unrelated background while preserving the similarity with the dog. This example too shows that by reducing the similarity of masked region with the unrelated known regions PAIntA enables prompt-faithful generation.

## APPENDIX E    DISCUSSION ON RASG

In this section we discuss the choice of RASG objective guidance function $S(x)$, then demonstrate the effect of RASG and motivate the part of gradient reweighting by its standard deviation.

### APPENDIX E.1    THE OBJECTIVE FUNCTION $S(x)$

As we already mentioned in the main paper, Stable Inpainting may fail to generate certain objects in the prompt, completely neglecting them in the process. We categorized these cases into two types, namely background and nearby object dominance issues. Chefer et al. (2023) also mentions these issues but for text-to-image generation task, and refers them as *catastrophic neglect* problem. To alleviate this problem Chefer et al. (2023) propose a mechanism called *generative semantic nursing*, allowing the users to "boost" certain tokens in the prompt, ensuring their generation. In essence the mechanism is a post-hoc guidance with a chosen objective function maximizing the maximal cross-attention score of the image with the token which should be "boosted". This approach can be easily adapted to the inpainting task by just restricting the maximum to be taken in an unknown region so that the object is generated there, and averaging the objectives across all tokens, since we don't have specific tokens to "boost", but rather care about all of them. In other words, by our notations from the main paper, the following guidance objective funciton can be used:

$$S(x_t) = -\frac{1}{|ind(\tau)|} \sum_{k \in ind(\tau)} \max_{i:\ M_i = 1} \{\overline{A}^k(x_t)_i\}. \tag{11}$$

However we noticed that with this approach the shapes/sizes of generated objects might not be sufficiently aligned with the shape/size of the input mask, which is often desirable for text-guided inpainting (see Fig. 10). Therefore, we utilize the segmentation property of cross-attention similarity maps, by so using *Binary Cross Entropy* as the energy function for guidance (see Eq. 9 in the main paper). As can be noticed from Fig. 10 the results with the binary cross-entropy better fit the shape of the inpaining mask.

### APPENDIX E.2    EFFECT OF RASG STRATEGY

Although the objective function $S(x)$ defined by Eq. 9 (main paper) results in better mask shape/size aligned inpainting, the vanilla post-hoc guidance may lead the latents to become out of their trained domain as also noted by Chefer et al. (2023): *"many updates of $x_t$ may lead to the latent becoming out-of-distribution, resulting in incoherent images"*. Due to this the post-hoc guidance mechanism

(semantic nursing) by Chefer et al. (2023) is done using multiple iterations of very small, iterative perturbations of $x_t$, which makes the process considerably slow. In addition, the generation can still fail if the iterative process exceeds the maximum iteration limit without reaching the necessary thresholds.

Thanks to RASG's seamless integration of the $\nabla_{x_t} S(x_t)$ gradient component into the general form of DDIM diffusion sampling, our RASG mechanism keeps the modified latents $x_t$ within the expected distribution, while introducing large enough perturbations to $x_t$ with only one iteration of guidance per time-step. This allows to generate the objects described in the prompts coherently with the known region without extra-cost of time.

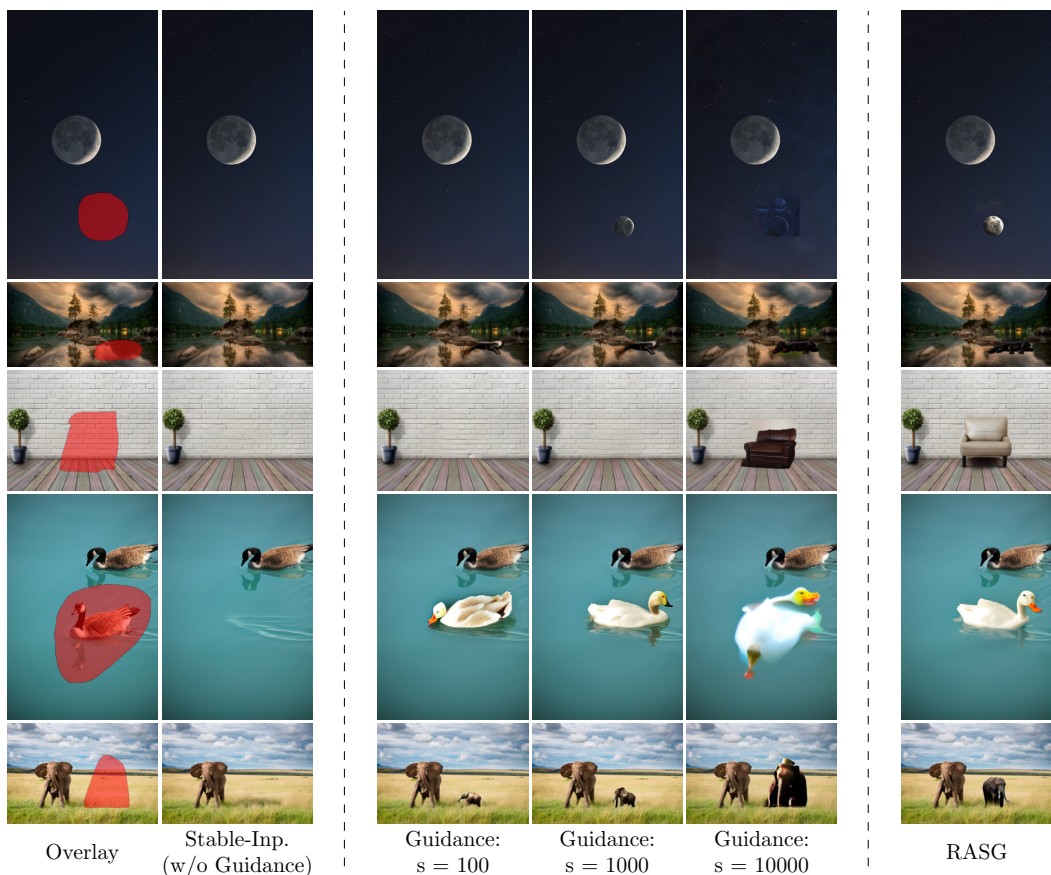

Figure 7: Comparison of RASG strategy with default Stable Inpainting and vanilla guidance mechanism with different guidance scales. In contrast to vanilla guidance, where the generation highly depends on the guidance scale, RASG consistently produces naturally looking and prompt-aligned results.

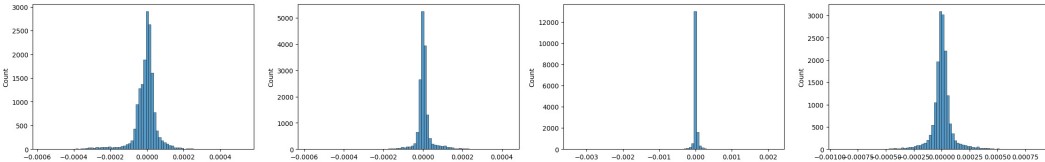

Figure 8: Histogram of $\nabla_{x_t} S(x_t)$ values (i.e. before gradient standardization)

Fig. 7 demonstrates the advantage of RASG's strategy over the vanilla guidance mechanism. Indeed, in the vanilla post-hoc guidance there is a hyperparameter $s$ controlling the amount of guidance. When $s$ is too small (e.g. close to 0 or for some cases $s = 100$) the vanilla guidance mechanism does not show much effect due to too small guidance from $s\nabla_{x_t} S(x_t)$. Then with increas-

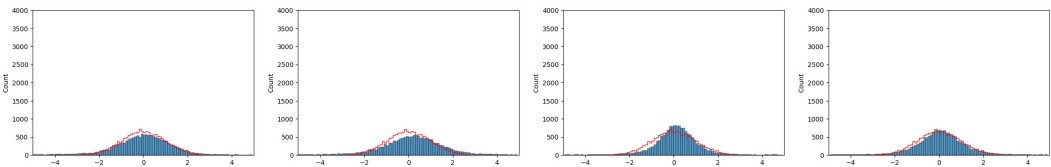

Figure 9: Histogram of $\frac{\nabla_{x_t} S(x_t)}{std(\nabla_{x_t} S(x_t))}$ values (i.e. after gradient standardization)

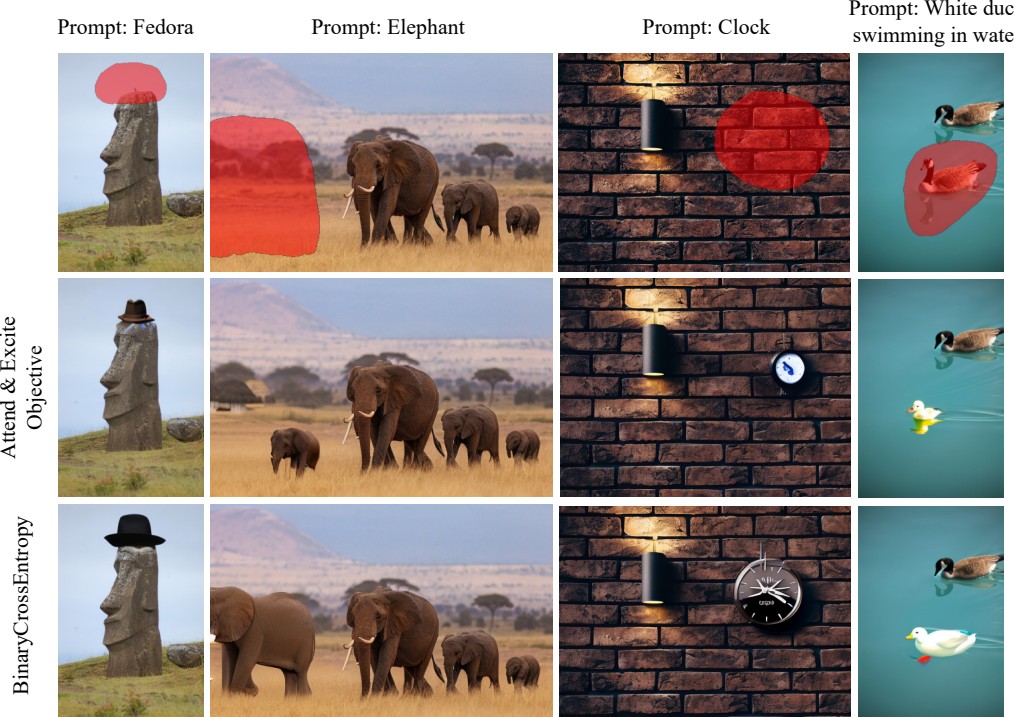

Figure 10: Comparison of the Binary Cross Entropy engery function to modifed version of Attend & Excite. Images generated from the same seed.

ing the hyperparameter ($s = 1000, 10000$) one can notice more and more text/shape alignment with prompt/inpainting mask, however the generated results are unnatural and incoherent with the known region. This is made particularly challenging by the fact, that different images, or even different starting seeds with the same input image might require different values of the perturbation strength to achieve the best result. In contrast, RASG approach is *hyperparameter-free* allowing both: prompt/mask-aligned and naturally looking results.

### APPENDIX E.3    RESCALING WITH STANDARD DEVIATION

The core idea of RASG is to automatically scale perturbation using certain heuristics, such that the guidance process has a consistent effect on the output, without harming the quality of the image. Our main heuristic relies on the fact that Song et al. (2021) have defined a parametric family of stochastic denoising processes, which can all be trained using the same training objective as DDPM Ho et al. (2020). Recall the general form of parametric family of DDIM sampling processes:

$$x_{t-1} = \sqrt{\alpha_{t-1}}\frac{x_t - \sqrt{1-\alpha_t}\epsilon_\theta^t(x_t)}{\sqrt{\alpha_t}} + \sqrt{1-\alpha_{t-1}-\sigma_t^2}\epsilon_\theta^t(x_t) + \sigma_t\epsilon_t, \tag{12}$$

where $\epsilon_t \sim \mathcal{N}(\mathbf{0}, \mathbf{1})$. Particularly $\epsilon_t$ can be taken to be collinear with the gradient $\nabla_{x_t} S(x_t)$ which will result in $x_{t-1}$ distribution preservation by at the same time guiding the generation process towards minimization of $S(x_t)$.

Therefore we propose to scale the gradient $\nabla_{x_t} S(x_t)$ with a value $\lambda$ and use instead of $\epsilon_t$ in the general form of DDIM. To determine $\lambda$ we analyse the distribution of $\nabla_{x_t} S(x_t)$ and found out that the values of the gradients have a distribution very close to a gaussian distribution, with $0$ mean and some arbitary $\sigma$, which changes over time-step/image (Fig. 8). Therefore, computing the standard deviation of the values of $\nabla_{x_t} S(x_t)$, and normalizing it by $\lambda = \frac{1}{std(\nabla_{x_t} S(x_t))}$ results in the standard normal distribution (see Fig. 9). So the final form of RASG guidance strategy is

$$x_{t-1} = \sqrt{\alpha_{t-1}} \frac{x_t - \sqrt{1 - \alpha_t} \epsilon_\theta^t(x_t)}{\sqrt{\alpha_t}} + \sqrt{1 - \alpha_{t-1} - \sigma_t^2} \epsilon_\theta^t(x_t) + \sigma_t \frac{\nabla_{x_t} S(x_t)}{std(\nabla_{x_t} S(x_t))}. \quad (13)$$

## APPENDIX F    VISUAL EXAMPLES FOR ABLATION STUDY

In Fig. 22 we show that RASG and PAIntA are crucial parts of our method. It's worth noting that for example in the cases when another instance of the prompt object is present in the known region (e.g. in the first row of Fig. 22 another zebra is present in the known part) PAIntA is still able to generate the desired object. This is thanks to its ability to enhance the impact of such known region pixels that are more aligned with the given prompt (zebra pixels in the known region will have high similarity to the prompt "zebra" resulting in high values of $c_j$).

However sometimes, especially in the case of *background dominance* issue, the PAIntA's ability to reduce the impact of the background non-prompt-related pixels is not enough (it does not allow the background to dominate through the attention scores, but also has no mechanism to directly force the inpainted region align the prompt), and, in such situations, RASG is helping to directly maximize the prompt-generation alignment in the missing region. Similarly, sometimes if only RASG applied, the background dominance can be so strong that RASG's direct guidance may be not enough for proper generation of the objects mentioned in the prompt. By so we conclude that the contributions of PAIntA and RASG are perpendicular and work best when applied together.

## APPENDIX G    GENERATION OF EXISTING OBJECTS

We noticed from Fig. 4 (main paper) that the competitor methods frequently fail to generate objects when the input image already contains an object of the same type. To verify that the failure is caused by the existance of the object, we first removed the existing object from the original and fill it in with background. For that we use the Stable Inpainting method, by providing an empty prompt. We then evaluate the competitor methods on the modified image with the original masks and prompts. As can be seen in Fig. 18, the models are in fact able to generate the corresponding objects in this case. We hypothesize that the reason of not generating the object when another instance of the same class is present is that cross-attention layers are designed to just ensure the presence of prompt objects without forcing them to be generated in a specific location indicated by the inpainting mask. In contrast with this our PAIntA layer forces the sampling process to pay more attention on the unknown region alignment with the prompt.

## APPENDIX H    MORE EXAMPLES OF OUR METHOD

We present more results of our method both for low-resolution (512 for the long side) images (Fig. 23), as well as high-resoltuion (2048 for the long side) (Figures 24, 25).

## APPENDIX I    LIMITATIONS

Although our method improves the prompt-alignment of existing text-guided inpainting approaches, it still has a dependency on the backbone model, hence inherits some quality limitations. Particularly it may generate extra limbs (the elephant in Fig. 11 has 5 legs) or illogical appearances (the sheep appears to have two bodies in Fig. 11 after the inpainting).

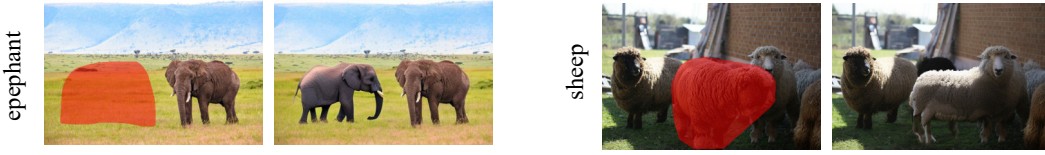

Figure 11: Failure examples produced by our approach.

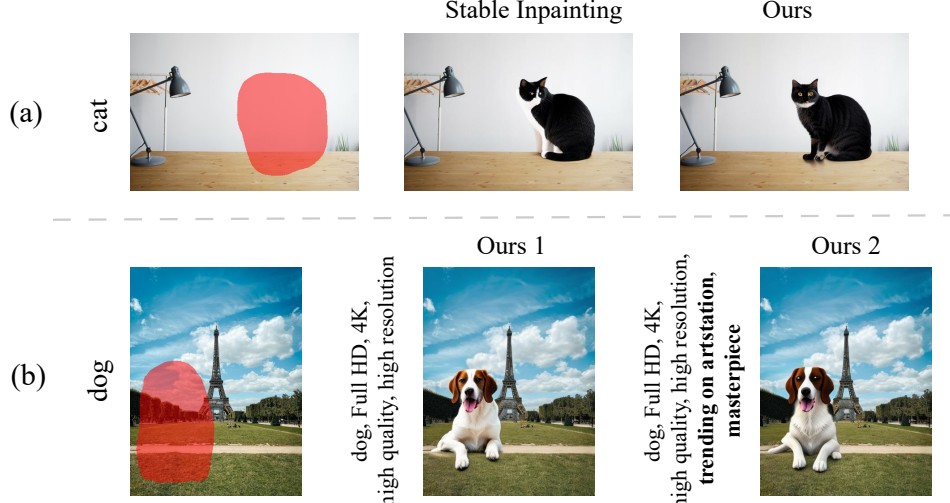

Figure 12: Sources of unnatural generation issues in our approach. (a) Unnatural generation case when the issue comes from the base model itself. (b) Example showing the sensitivity of our approach to carelessly chosen positive words. Left result is generated using only neutral positive words. Right result is generated using additional, non-aligned positive words causing generation to be less natural.

Besides illogical issues, our method can sometimes produce unnatural results. First of all, unnatural results can be attributed to the underlying base model, which in our case is Stable Inpainting (e.g. see Fig. 12.a). Furthermore, sometimes our approach, due to its prompt-faithfulness, may become over-sensitive to positive/negative words which may lead to unnatural results when those are not carefully picked. For example, as can be seen in Fig. 12.b, the addition of positive words *trending on artstation* and *masterpiece*, which usually appear in the context of artistic images, causes the generated dog to become considerably less natural as opposed to the generation with only neutral positive prompts. This happens, because our method limits the use of prompt-unrelated visual context, and guides the generation towards the user provided prompt, so that in case prompt contains unrelated positive tokens, the generated content may be shifted from the visual context.

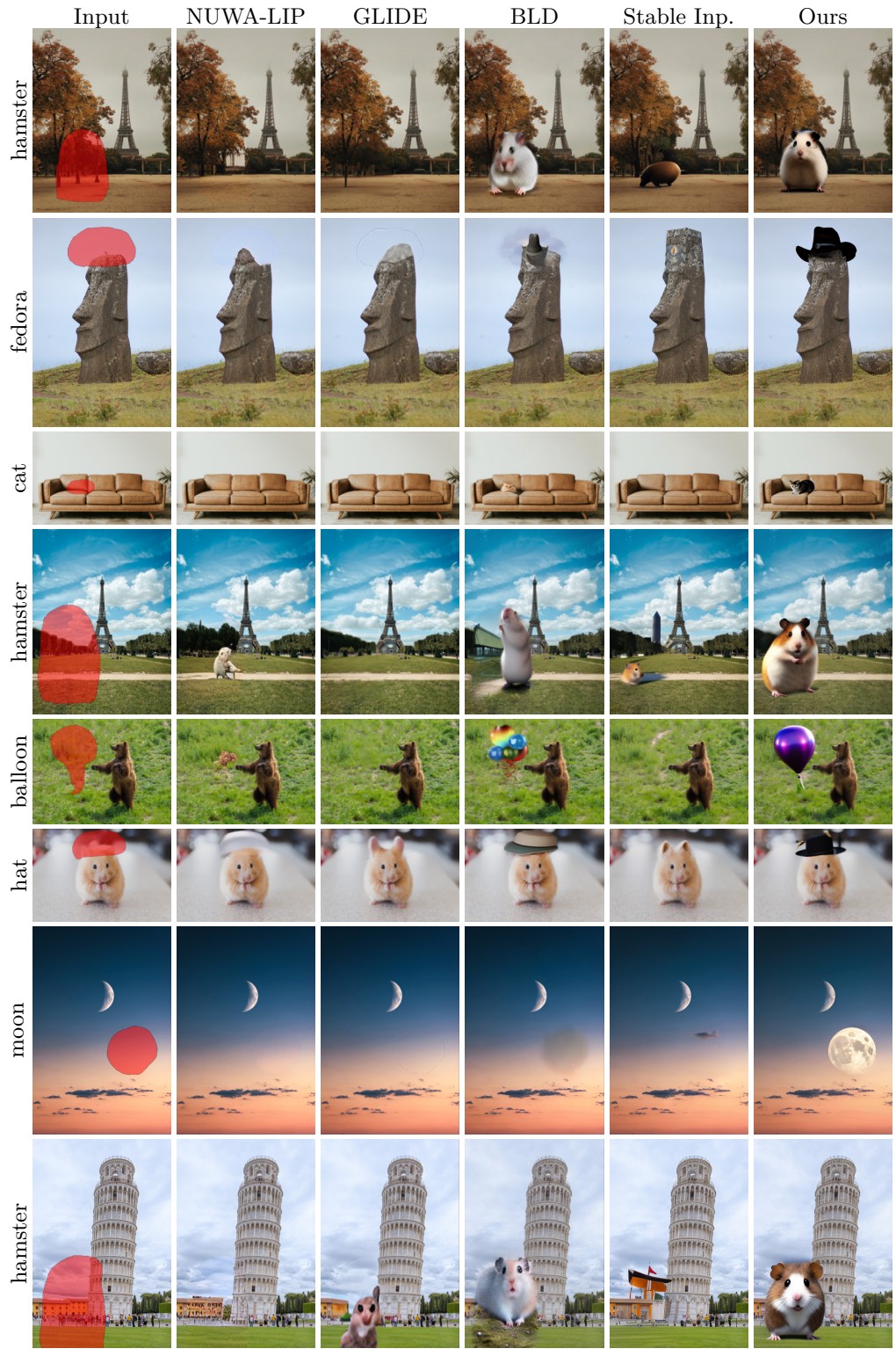

Figure 13: More qualitative comparison results.

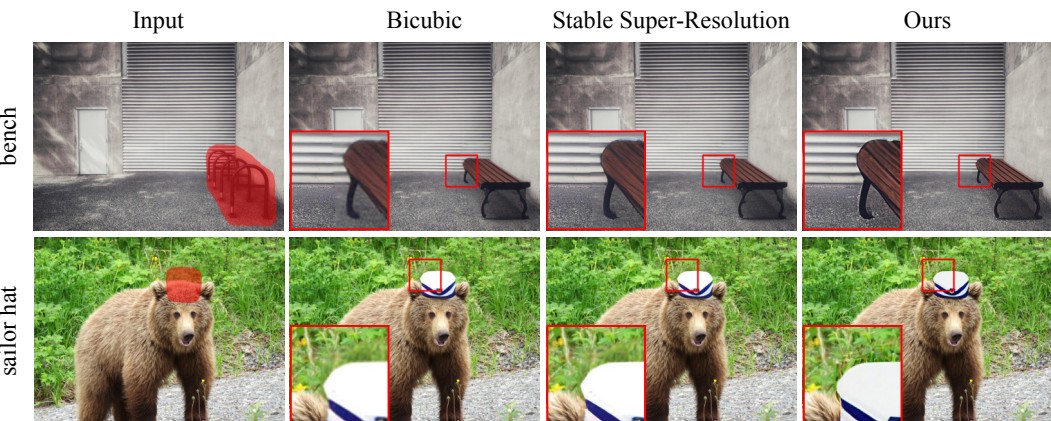

Figure 14: More qualitative comparison results on MSCOCO 2017.

Figure 15: Comparison of our inpainting-specialized super-resolution approach with vanilla upscaling methods for inpainting. Best viewed when zoomed in. For more results see Appendix.

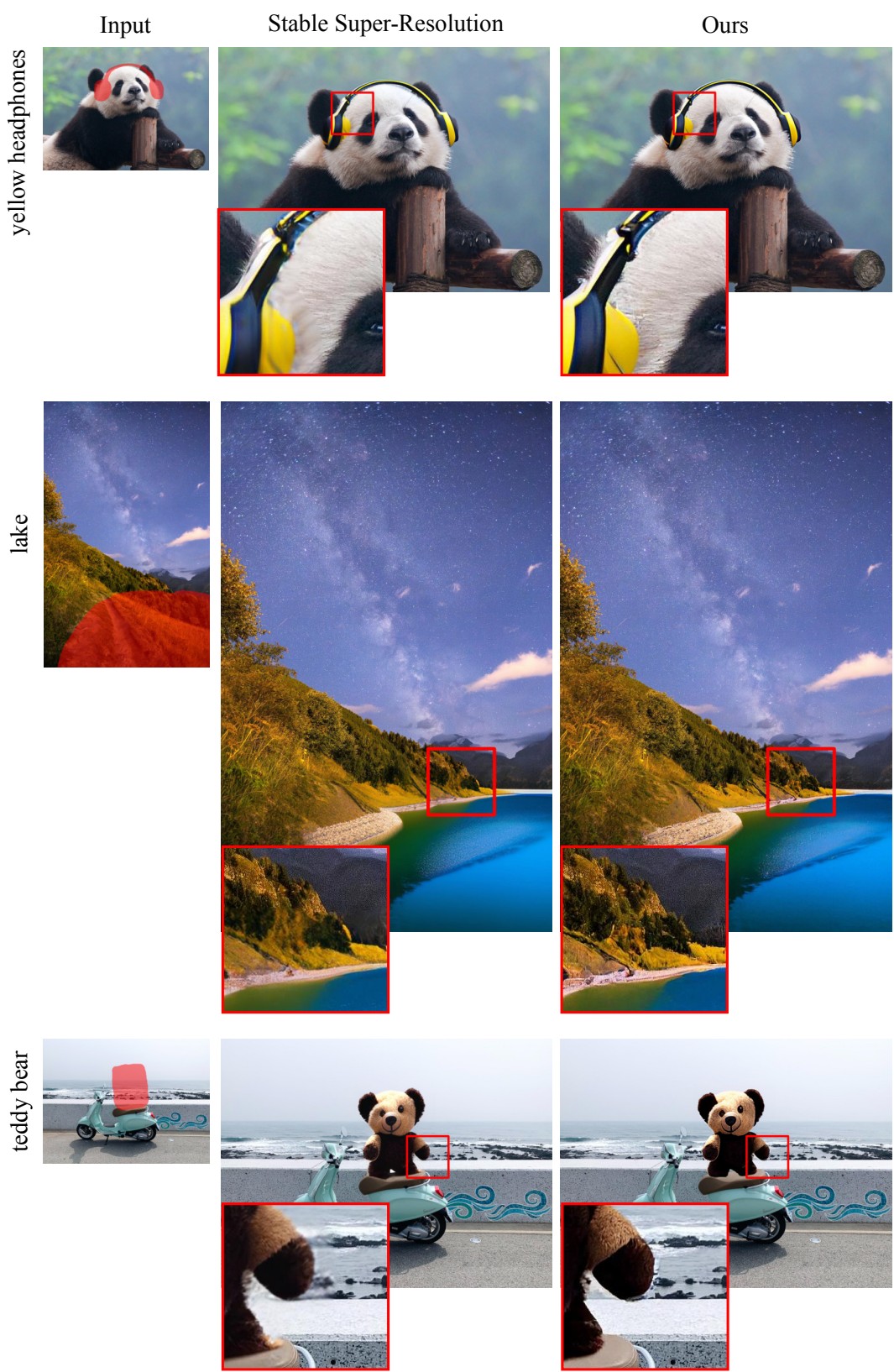

Figure 16: Comparison between vanilla SD 2.0 upscale and our approach. In all examples the large side is 2048px. The cropped region is 256x256px. Best viewed when zoomed in.

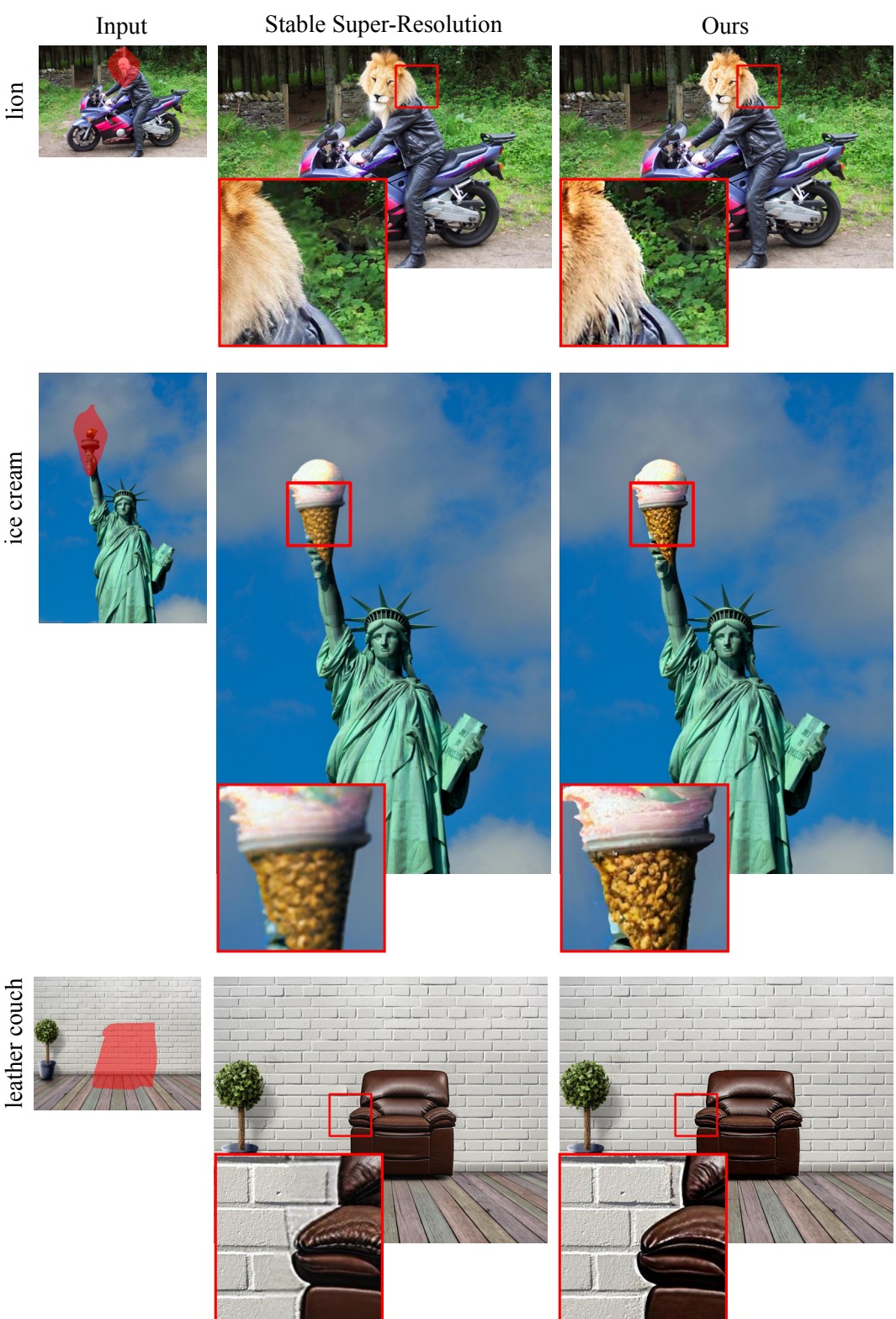

Figure 17: Comparison between vanilla SD 2.0 upscale and our approach. In all examples the large side is 2048px. The cropped region is 256x256px. Best viewed when zoomed in.

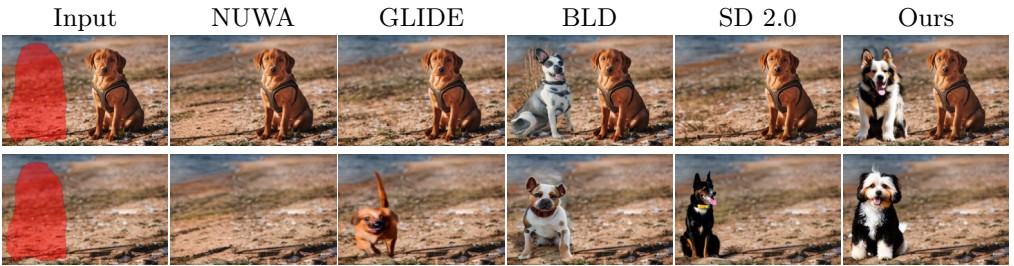

Figure 18: Competitor models have no issue generating the same object if an existing copy of it is removed

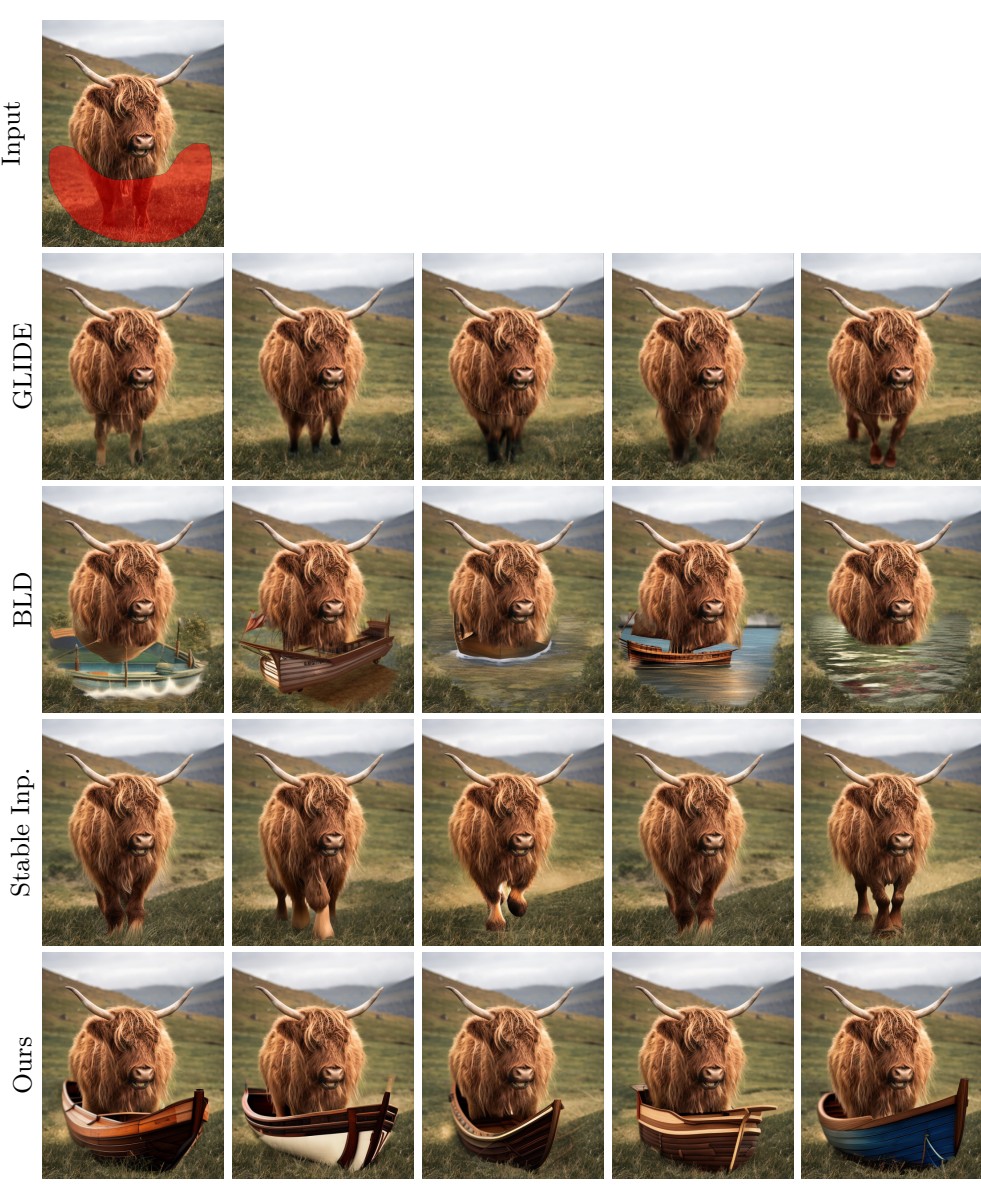

Figure 19: Comparison with baselines when generating more results from different seeds (1/3). Prompt: boat

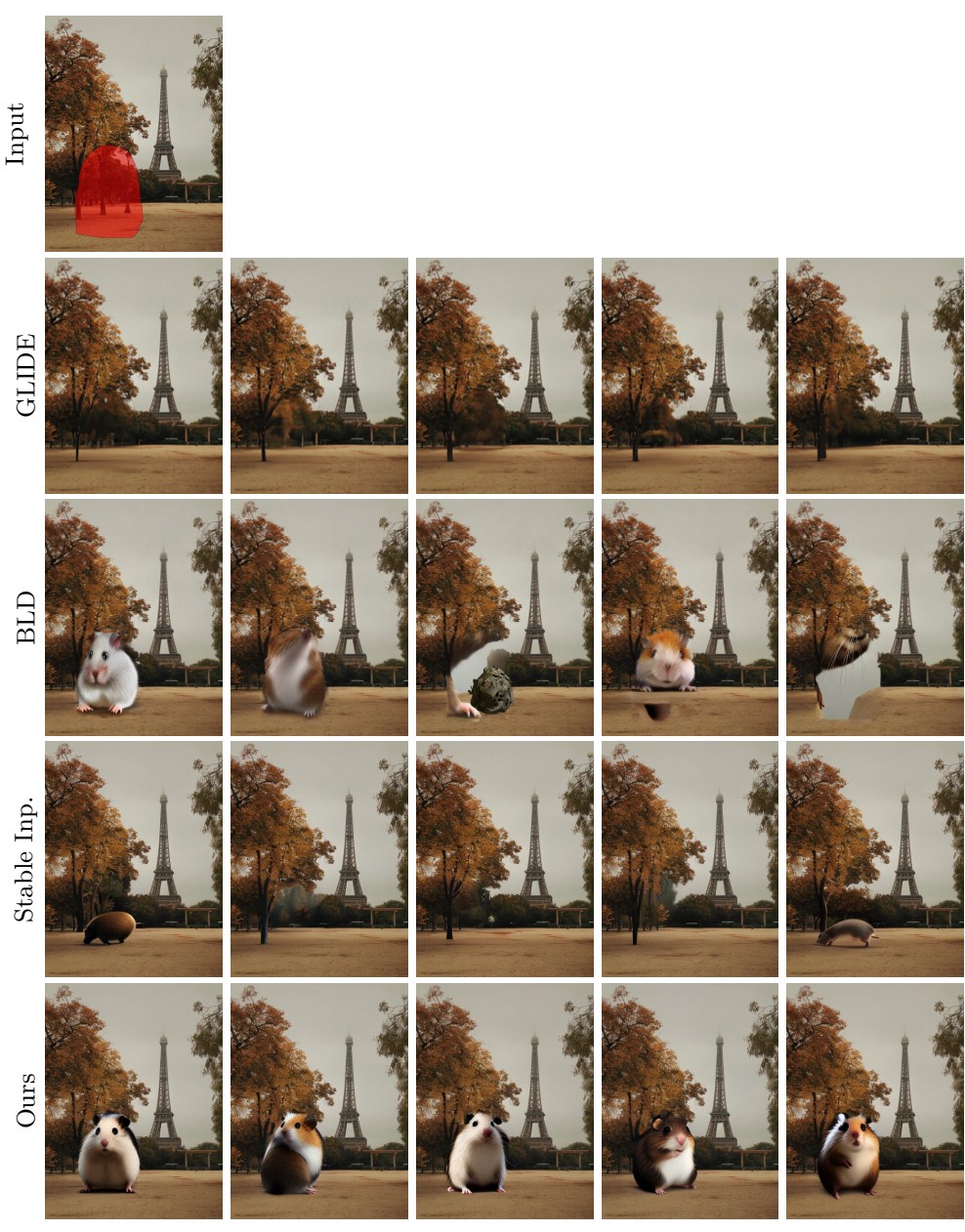

Figure 20: Comparison with baselines when generating more results from different seeds. (2/3). Prompt: hamster

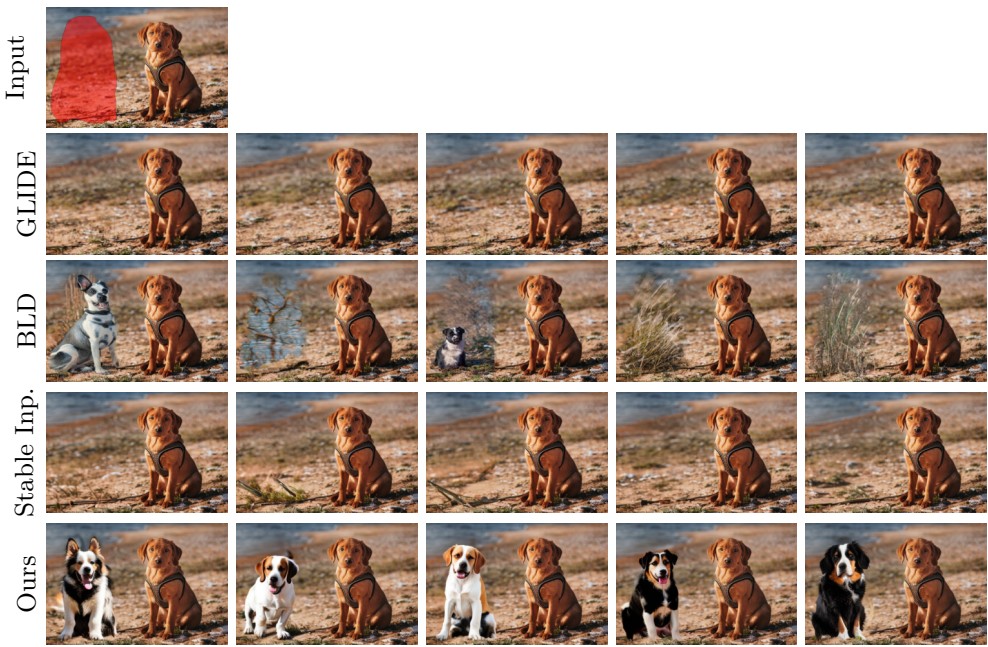

Figure 21: Comparison with baselines when generating more results from different seeds. (3/3). Prompt: dog

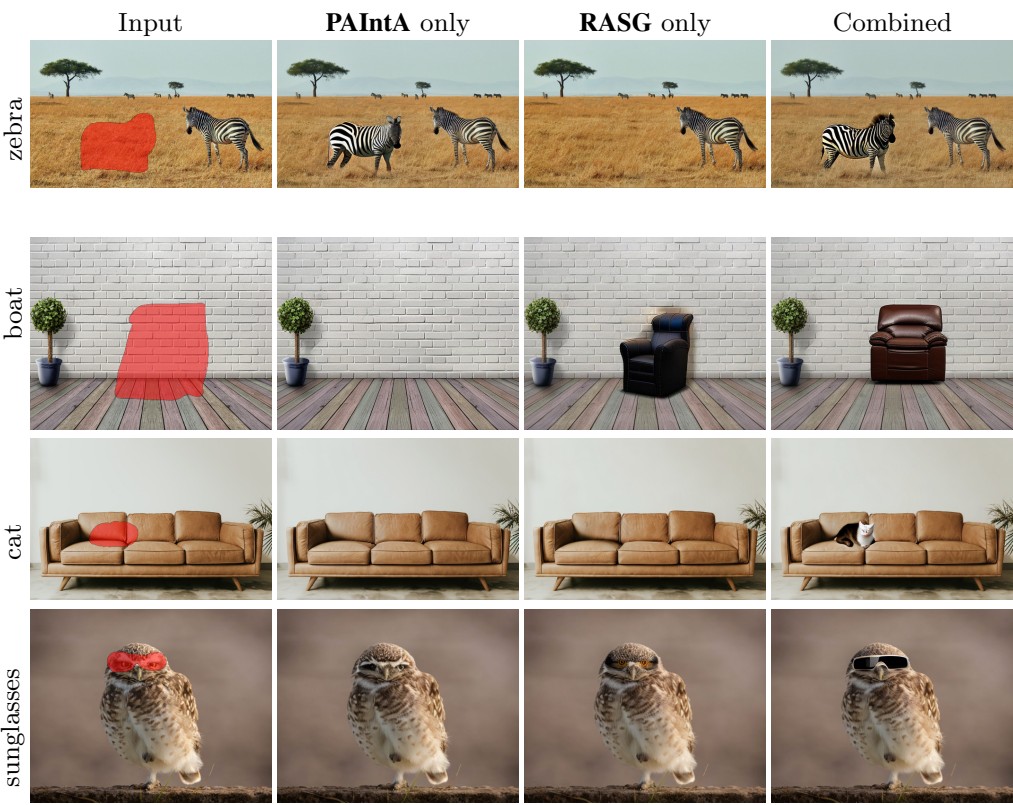

Figure 22: Ablation study of the visual results of our contributions RASG and PAIntA.

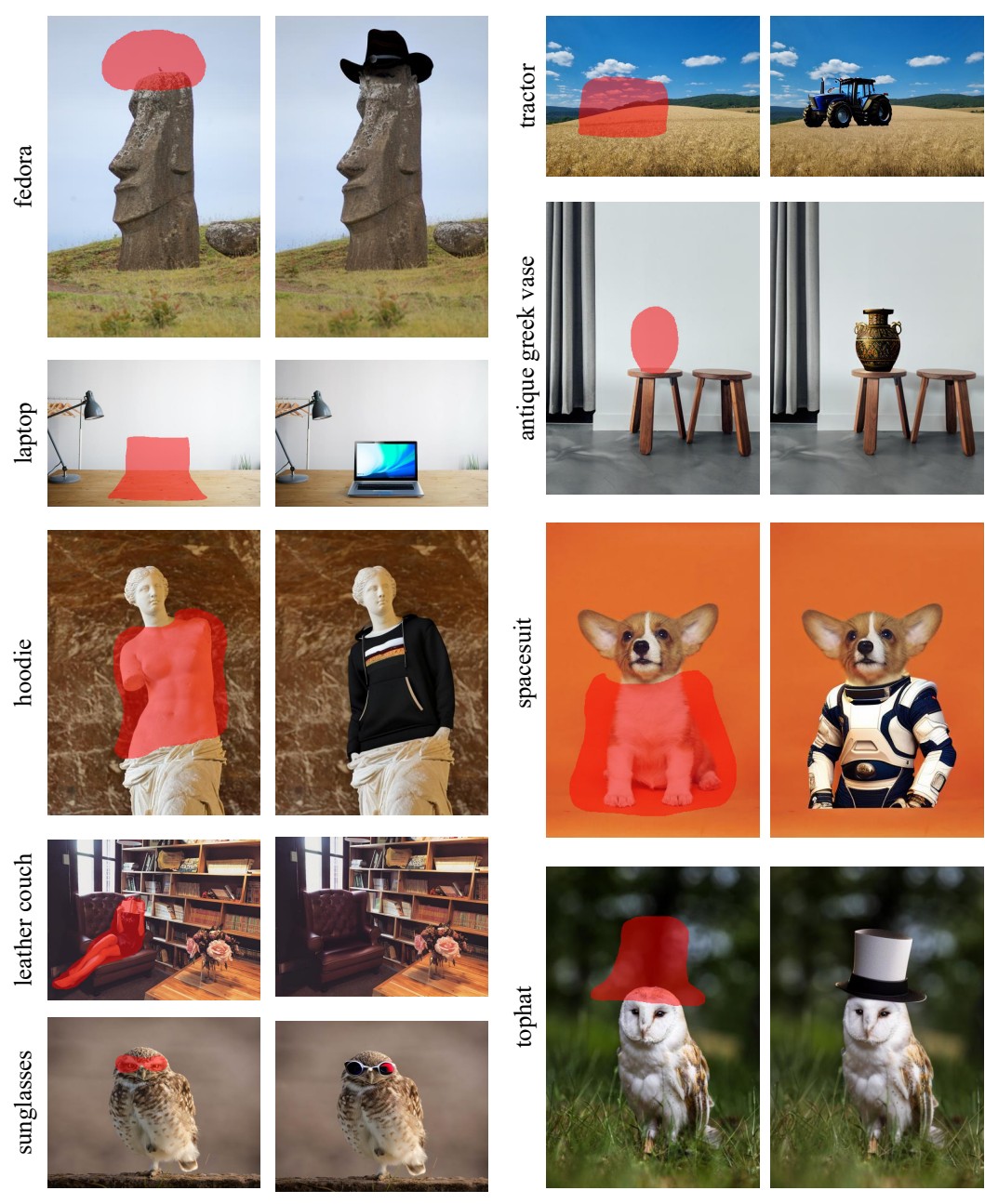

Figure 23: More results of our method.

boots

parrot

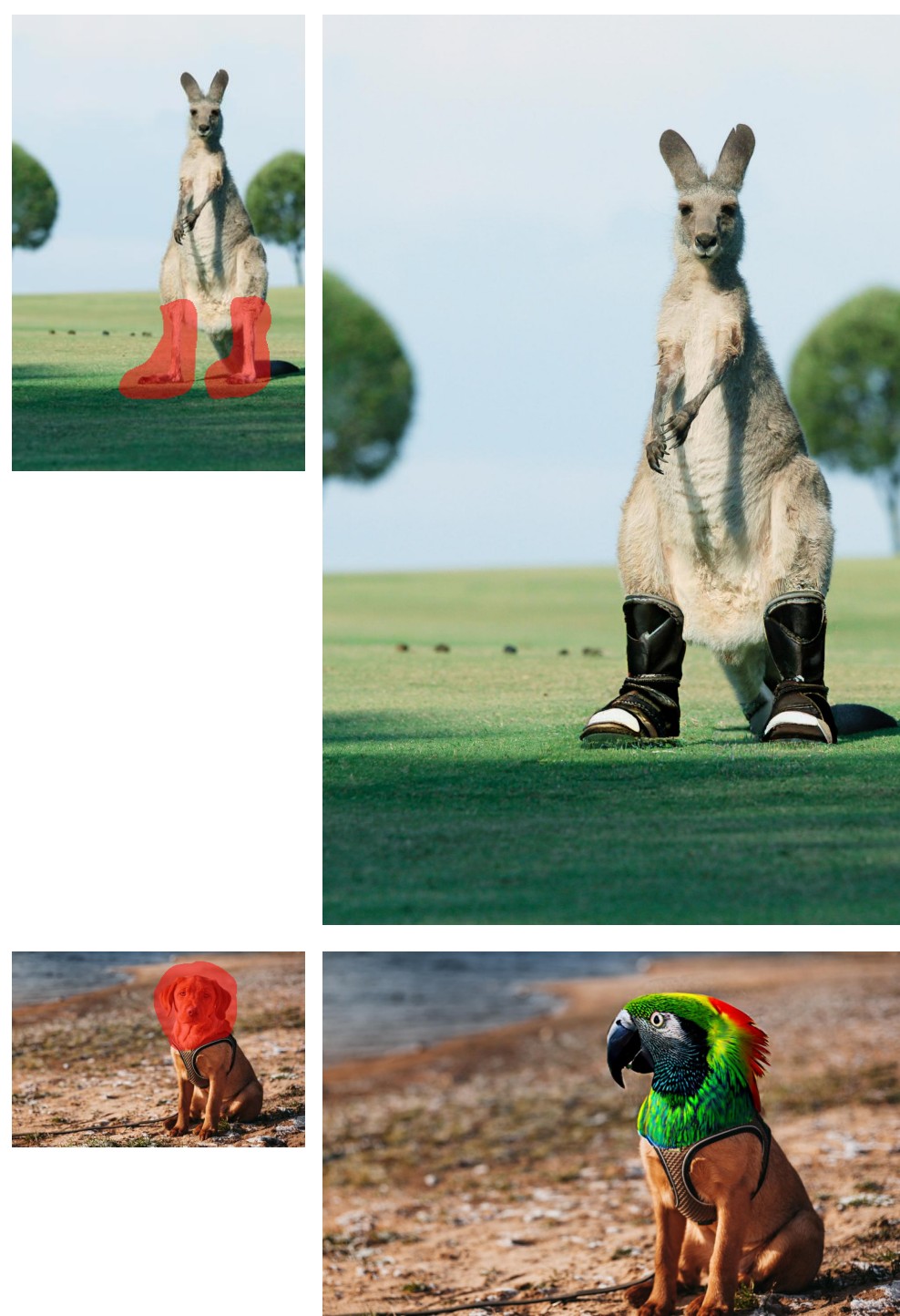

Figure 24: More high-resolution results of our method.

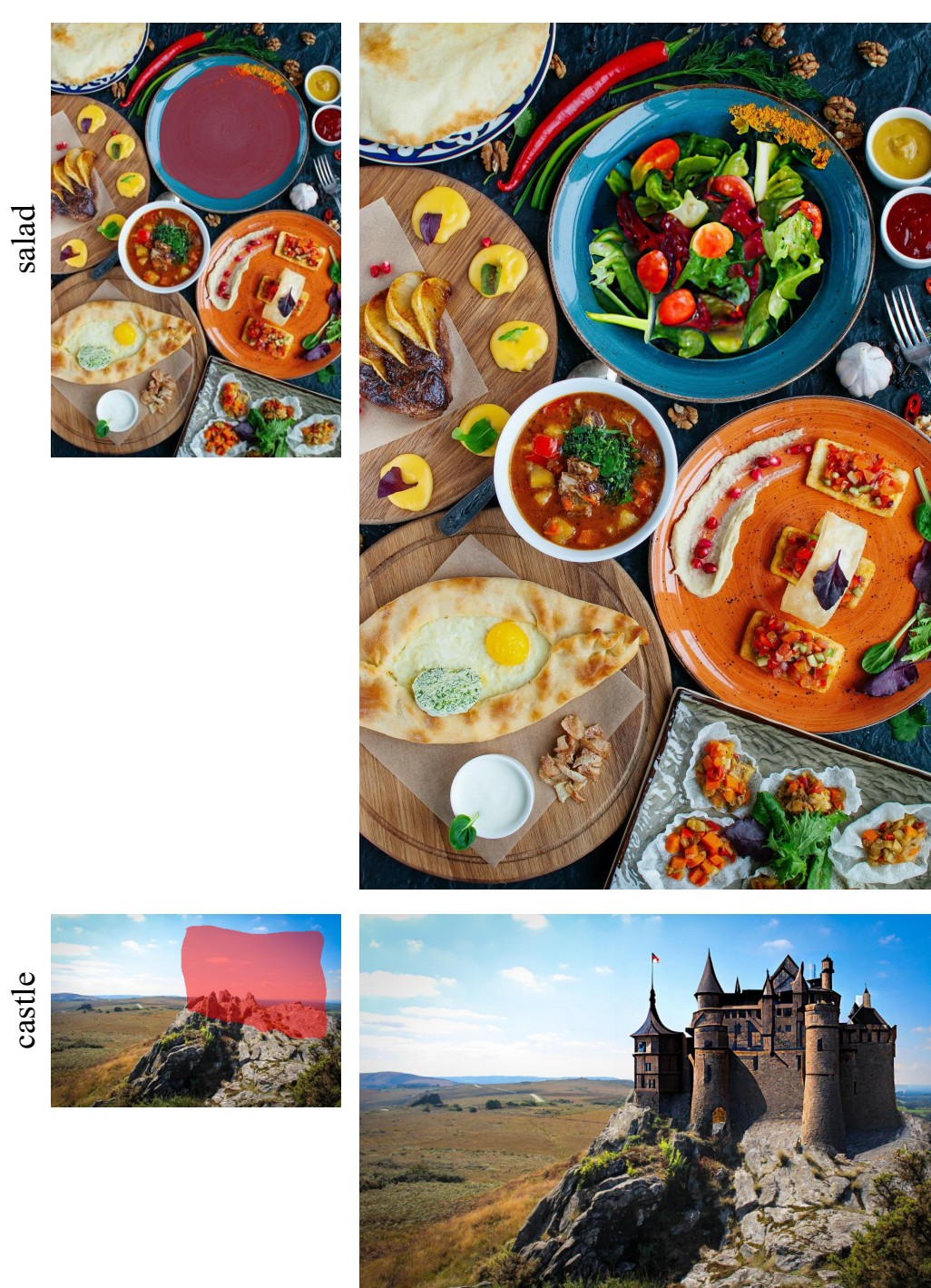

Figure 25: More high-resolution results of our method.