# OpenReview forum: "Fill with Anything: High-Resolution and Prompt-Faithful Image Completion"
_ICLR.cc/2024/Conference — ICLR 2024 Conference Withdrawn Submission_

### Official Review · Reviewer_WycU · 2023-10-17

**Soundness:** 3 good
**Presentation:** 2 fair
**Contribution:** 3 good
**Rating:** 5
**Confidence:** 4

**Summary:**

This paper proposed a training-free approach that faithfully guided the local image inpainting according to the textual prompts, which can be further scaled to high-resolution image inpainting. The authors proposed the Prompt-Aware Introverted Attention (PAIntA) layer to make self-attention modules focus more on text-related unmasked regions. Then, the Reweighting Attention Score Guidance (RASG) improved the textual alignment through cross-attention scores. Thus, this paper enjoys better performance with faithful generations.

**Strengths:**

1. Both PAINTA and RASG are convincing with clear motivations.
2. The guidance from cross-attention scores of RASG is interesting. And RASG also enjoys good quantitative improvements. But it misses some important discussions about the related works.
3. Some inpainting results in the supplementary are amazing.

**Weaknesses:**

1. The main concern is the implementation of PAINTA, which rudely adjusts the attention scaling for self-attention according to the cross-attention score. Many visualized results in both the main paper and the supplementary are over-saturated, which might be caused by the adjusted self-attention scale of PAINTA.  Whether the over-saturated generation related to PAINTA or self-attention scaling?
2. Although RASG is highly inspired by Chefer et al. (2023), the authors do not provide further clarification about the difference. Moreover, the effect of some details is not well verified, such as the std-based normalization of RASG. More related works about RASG should be considered.
3. Except for the over-saturated issue, the quantitative improvement of PAINTA is minor. Does this technique just work for some failure cases such as "zebra" in Fig13 of the supplementary?
4. More in-depth discussions about PAINTA should be discussed. Such as some visualizations about $c_j$ in some representative cases.
4. The contribution of the high-resolution inpainting is minor, which should be considered as an implemented trick rather than a main contribution.

**Questions:**

1. It would be interesting if the authors could discuss more details about weaknesses2.
2. The qualitative result of "cat" in Fig.13 is a little confusing. Both "PAIntA only" and "RASG only" fail to generate the target in the masked region. But the combined one successfully synthesizes a cat on the sofa.

---

> ### Author Response · Authors · 2023-11-19
> **Thank you for the detailed review! (1/2)**
>
> We appreciate the reviewer's valuable comments and recognition of our paper's strengths such as PAIntA and RASG clear motivation, RASG benefiting from binary cross-entropy guidance with cross-attention scores, and high quality results of our method. Below we address the questions/weaknesses the reviewer pointed out. Due to space limitations we split our response into two comments.
>
> **A1** We understand the concern, and we would like to address it in two parts. Firstly, we would like to highlight that to make the PAIntA’s contribution and its effect more clear we added a new section in our revised supplementary material solely devoted to more discussion on PAIntA (Appendix D). We kindly ask the reviewer to take a look. There we have presented a detailed analysis of the effect the rescaling of PAIntA has on the final outcome, particularly Fig. 6 in Appendix demonstrates that PAIntA’s changes are able to inpaint various images without introducing any unnecessary artifacts.
>
> Second, our contributions PAIntA and RASG, are plug-and-play, hence our method may inherit quality limitations of its base model, i.e. Stable Inpainting. We noticed that Stable Inpainting itself sometimes generates unnatural and over-saturated results (see Fig. 12 (a), Appendix).
>
> Additionally, when investigating the reason for some over-saturated results, we noticed that our method is slightly more sensitive to the positive/negative prompt choice. Therefore we have reviewed the list of positive and negative prompts used in our examples, and noticed that the issue is mostly caused by some specific positive prompts we use (like “masterpiece and trending on artstation”) which are helping to improve the generation in general but in some cases cause a slight over-saturated effect (see Fig. 12 (b), Appendix). To communicate this and other limitations more clear we added a new Limitations section in our supplementary material (Appendix I). Fortunately, the over-saturation issue in our examples is not crucial to the main talking points we wanted to discuss in the paper and can be significantly alleviated by choosing the positive/negative prompts.
>
> **A2** Following this and also other comments, we added a new section in our revised supplementary material devoted to more discussion on RASG (Appendix E). We kindly ask the reviewer to take a look.
>
> While RASG is inspired from the work by Chefer et al. (2023) it has 2 major differences:
>
> 1. We use a different objective for the post-hoc guidance technique, namely cross-entropy between the inpainting and the cross-attention masks. This gives us the opportunity to guide the diffusion backward process towards generating objects better aligned to the given mask size/shape. We show the effect of such a guidance over the guidance objective from Chefer et al. (2023) in Appendix E.1 (see Fig. 10)
> 2. Performing vanilla post-hoc guidance techniques usually introduces a distribution shift in the latents (as also noted by Chefer et al.), due to the additional gradient term. Due to this Chefer et al. perform the guidance by multiple iterative latent perturbations which makes the process considerably slow. In contrast, our RASG strategy seamlessly integrates the gradient term into the general form of DDIM sampling process resulting in prompt-faithful and in-domain outputs. We show the effect of RASG sampling strategy in the revised version of Appendix E.2.
>
> In addition to that we performed an analysis on the std-based normalization of RASG and presented it in our revised supplementary material, Appendix E.3.
>
> Additionally we found the suggestion of the reviewer to add more related work on post-hoc guidance mechanisms extremely useful, so we dedicate a subsection in our Related Work section.
>
> **A3** While we discuss the over-saturation issue in our Limitation section, we respectfully disagree that PAIntA’s quantitative improvement is minor. Especially for CLIP-score we get 25.24 vs 24.86 which is a significant difference when taking into account the specifics of CLIP-score (it is very sensitive, its range is not too large, roughly from 24.0 to 27.0).
>
> For more analysis on PAIntA we added a new section in the revised Appendix (Sec. D) and show the impact of PAIntA layer (Fig. 6, Appendix). The figure demonstrates 6 examples where PAIntA helps to solve two issues, namely background dominance (rows 1, 2, 3) and nearby object dominance (rows 4, 5, 6).
>
> We can clearly see that PAIntA is able to solve not only cases such as “zebra”, when there is another instance of the same class in the known regions, but also generate objects not present in the image.

---

> ### Author Response · Authors · 2023-11-19
> **Thank you for the detailed review! (2/2)**
>
> **A4** We thank the reviewer for the suggestion and the provided interest. Originally we have tried to focus on the final outcomes of our proposed techniques, and avoid overburdening the paper with all our performed experiments.
> However, seeing that the work principles of PAIntA have stricken the interest of most reviewers, we gladly share some in-depth analysis and visualizations regarding its working principles. The analysis can be found in PAIntA Discussion section in our revised Supplementary material (Sec. D). We hope that you will find these new visualizations interesting.
>
> **A5** We respectfully disagree with the reviewer concerning the minor contribution of  our inpainting specialized super-resolution. We believe that this simple yet effective technology can be of a great help to the inpainting community, and its technical simplicity should not downplay the importance of its contribution. We find that the introduction of this technique fills in an important gap in the existing methods. While similar techniques have been used in the community for other reasons, like image-inpainting, to the best of our knowledge no such thing was proposed for obtaining a meaningful high-resolution inpainting pipeline utilizing the high frequencies of the known region.
>
> With the existing techniques, the users either have to upscale an entirely generated image and lose the original high-frequency details of the original image even in the known region (and typically obtain a lower known region). Alternatively, the upscaled image can be blended with the original image post factum, which is bound to introduce visual artifacts around in the seams region, and in general will have a masked region that is less correlated to the known region (see Figures 15, 16, 17 in Appendix). To the best of our knowledge we are the first to propose a technique that allows the seamless blending of generated objects with high-resolution input images. Needless to say that this technique is also plug-and-play, and can be used as a convenient baseline for future study on this problem.
>
> In summary we found it important to mention it in the introduction of our paper, since we believe that it can have a significant impact, despite its technical simplicity. In addition, it can be swiftly integrated in many existing techniques and prove to be useful to the community. However, prompted by this comment we decided to rewrite our contribution points, to tone down this contribution w.r.t the other two. We hope that this new wording is better aligned and better represents our view regarding this contribution.
>
> *Concerning the questions the reviewer asked:*
>
> 1. We tried our best to address the weaknesses mentioned by the review and hope our comments introduce a sufficient clarity and highlights the whole picture of contributions of this work.
> 2. We understand the confusion coming from our ablation image in the Appendix, hence we added more discussion in Appendix F describing the reasons why their contributions/improvements are complementary to each other hence they work best when combined. In summary:
>     1. PAIntA enhances the self-attention scores to gain more prompt faithful results. However the attention score enhancement is designed to only prevent the background or nearby object dominance over the prompt when generating the missing region. Sometimes just eliminating the undesired influence of the known region is not enough for prompt-aligned generations, hence we also leverage a post-hoc guidance mechanism such as RASG.
>     2. RASG applies a post-hoc guidance to improve the text alignment even further, however to avoid undesired latent distribution shifts it introduces the strategy of combining the guidance with the general form of DDIM process. Using only RASG may also lead to non prompt-aligned results since the undesired influence of the known region (through self-attention layers) may be so strong that even directly guiding the generation via RASG will fail to generate the prompt objects.
>
>     Therefore we conclude that the contributions of PAIntA and RASG are complementary to each other and best used in combination

---

> ### Comment · Reviewer_WycU · 2023-11-22
> **Thanks for the response from the authors**
>
> Thanks for the response from the authors.
>
> The response addresses some of my concerns.
> As mentioned by Reviewer Q343, the implementation of PAINTA is theoretically not perfect, which weakens the contribution of this work.
> Moreover, this paper has missed many important discussions about related works. Though the authors have made efforts to address this in the revision, the added content seems to have significantly compressed the official ICLR layout. Therefore, a major revision is recommended to better organize and enhance both the discussions on related works and the methodological introduction to improve the overall quality of this paper.
>
> Besides, the authors did not show the visualization of $c_j$, while only self-attention maps are provided in the revision.

---

### Official Review · Reviewer_Q343 · 2023-10-18

**Soundness:** 3 good
**Presentation:** 3 good
**Contribution:** 3 good
**Rating:** 3
**Confidence:** 5

**Summary:**

This paper introduces modifications to enhance the text-guided inpainting performance of the stable diffusion inpainting model without additional training. The proposed enhancements involve adjustments to the attention scores within self-attention layers and the incorporation of classifier guidance through cross-attention scores. Additionally, refinements are made to the super-resolution model to improve image completion.

**Strengths:**

This paper demonstrates effectiveness when compared to the standard stable diffusion model, suggesting potential capacity improvement without the need for retraining.

**Weaknesses:**

This paper contains architectural design flaws and lacks experimental validation for each modification. Furthermore, it overlooks significant related research.

**Questions:**

1. The proposed PAIntA method claims to "mitigate the too strong influence of the known region over the unknown by adjusting the attention scores of known pixels contributing to the inpainted region". However, it remains unclear whether the $c_J\in (0,1)$ parameter effectively reduces this influence, especially considering that the element values of the similarity matrix $A_{self}$ are not always positive. Further discussion is needed to validate the effectiveness of the proposed PAIntA method.

2. The proposed RASG technique appears to bear a resemblance to the work by Chefer et al. (2023). The authors assert the superiority of RASG without presenting theoretical or empirical evidence. Some supporting evidence or comparisons are necessary to substantiate this claim.

3. Since RASG is a classifier guidance technique, it is essential to provide a comprehensive discussion of related works to distinguish RASG from other similar methods. This would help readers better understand its unique contributions and advantages.

4. It is not clear why RASG and PAIntA are applied at different resolutions. Specifically, the rationale for not using RASG in the $H/16\times H/16$ resolution should be explained.


5. Using Poisson blending needs clarification. How do you prevent the original image information from leaking into the inpainting region?

---

> ### Author Response · Authors · 2023-11-19
> **Thank you for the detailed response!  (1/2)**
>
> We thank the reviewer for the huge work done, and appreciate they emphasize the advantage of our method over SD, as well as the training-free nature of our approach. Below we address the questions the reviewer asked in the review. Due to space limitations we address the questions in two comments.
>
> **A1** We thank the reviewer for this valuable comment. We added more discussion on PAIntA (Appendix D) in the supplementary material of the revised version where we visualize how effectively $c_j$ rescaling reduces the influence of the known region (Fig. 6, Appendix). The results clearly show that in the case of vanilla self-attention the inpainted region is mostly affected (has much higher attention scores with) by the **undesired parts** (not aligned with the given prompt) of the known region. However after adding the PAIntA block (i.e. after adjusting the attention scores by $c_j$ prompt-alignments) we can see that 1) the highest attention scores of the inpainted region are with itself (which in its turn is affected by the textual prompt via cross-attention layers, so will tend to generate prompt-aligned results); 2) the attention scores with prompt-aligned known region pixels are higher than the ones which are not aligned (the examples of the dog, elephant, or zebra when we generate the second object of an existing type).
>
> Concerning scaling non-positive attention scores with $c_j\in (0,1)$ we apologize and understand the confusion: When ($A\_{self})\_{ij}<0$, and assume $c_j$ is close to zero (i.e. the pixel $j$ should not contribute to forming the pixel $i$) then $c_j(A\_{self})\_{ij}$ will increase the attention score instead of decreasing it. However $c_j(A\_{self})\_{ij}$ is still lower than zero, whereas the attention scores of the inpaining region with itself turn out to be significantly higher than zero (see Fig. 6, Appendix, we hypothesize this is because they are similar each other and are not scaled by $c_j$), by so taking the large amount of contribution after applying SoftMax even if we unnecessarily slightly increase some attention scores of the known region over the inpainted region.
>
> **A2** We thank the reviewer for pointing this out. As a response, we have added a new section solely for the discussion of RASG to our supplementary material (Appendix E) and kindly ask the reviewer to take a look.
> Nevertheless we would like to rectify what we think was poorly communicated by us beforehand.
>
> Indeed, the work of Cheref et al. discusses the issue of “catastrophic prompt neglect” which bears a significant resemblance to our discussions of “prompt neglect” (in the new revision), and we do not hide that RASG was partially inspired by their work. However, we would like to note that the issues discussed in these papers are discussed in significantly different contexts: namely image generation and image completion, which leads to major differences.
>
> The work by Chefer et al. is tuned to solve the issue of objects neglected during image generation. There is no suggestion about any spatial locations the object has to appear in, and there are no pre-existing surroundings the object should fit into. Due to this, the techniques of Chefer et al. are designed to ensure **the existence** of the described objects.
>
> In our case, the positioning of the object and its shape are also important. With this in mind, the work of Chefer et al. is not directly comparable to ours, since the way it is, without any modification, it is not a solution for our presented problem. Due to this, the claim of RASG’s superiority was directed not to Chefer et al., but rather to the **vanilla post-hoc guidance** method to alleviate the “prompt neglect” issue with a strategy **preventing the latents becoming out-of-distribution** due to gradient shifts. We added a discussion and grounding of this superiority in Appendix E.2 (see Fig. 7)
>
> Although the work of Chefer et al. is not a direct competitor of RASG, its guidance objective function can be modified to guide the inpainting diffusion process as well. To this end we compare (see Fig. 10, Appendix) their objective function with the one we introduce (leveraging the open-vocabulary segmentation property of cross-attention layers) in the scope of RASG and show the advantage of our design choice of the function (for more details please see Appendix E.1).
>
> **A3** We thank the reviewer for pointing this out. Due to tight space constraints we had to evaluate which sections and topics would make it to the main paper text. Reflecting on the feedback received from the reviews, especially this question, we have performed some structural changes to our Related Work section, dedicating some discussion to post-hoc guidance techniques. We kindly ask the reviewer to take a look and we hope we made the contribution of RASG more clear.

---

> ### Author Response · Authors · 2023-11-19
> **Thank you for the detailed response! (2/2)**
>
> **A4** This choice was motivated mainly by performance reasons. Compared to PAIntA’s rescaling, the gradient computations included in RASG are computationally more expensive. For PAIntA, we chose the resolutions that provided the best qualitative results. For RASG, however, we found that the additional computational overhead introduced by the H/16×H/16 resolution was not worth the improvements, which were neglectable in comparison. To not overburden the paper with all our unsuccessful experiments, we decided to omit this version from the paper.
>
> **A5** We apologize for the confusion. When applying the Poisson blending we use a slightly dilated mask and the seamlessClone() function from OpenCV with the NORMAL_CLONE flag. This algorithm focuses on copying the source image into the target image while also harmonizing the source and the target images in the boundary region of the mask. This allows to seamlessly blend the generated region into the original image without having the original image information impact the result.

---

> ### Comment · Reviewer_Q343 · 2023-11-22
> **Thanks for the response**
>
> Having considered the responses from the authors, I still uphold my initial decision. I suggest that the authors undertake a thorough revision of the paper to improve its overall clarity and readability, and modify the methodology to avoid mistakes in the proposed algorithm.

---

### Official Review · Reviewer_ZNkY · 2023-10-21

**Soundness:** 3 good
**Presentation:** 3 good
**Contribution:** 3 good
**Rating:** 6
**Confidence:** 4

**Summary:**

This paper introduces a novel training-free method for text-based image inpainting task, which is built upon pre-trained StableDiffusion inpainting and super-resolution models. The proposed method has two technical contributions: PromptAware Introverted Attention (PAIntA) and Reweighting Attention Score Guidance (RASG). The experiment metrics and visual results demonstrate the superiority of the proposed method over other state-of-the-art methods.

**Strengths:**

- This work aims to address the text-based image inpainting task, which is useful for practical applications. Prior methods suffer from unstable inpainting results that may lack text-image alignment and good quality/resolution. This work proposes a two-stage pipeline built upon pre-trained StableDiffusion inpainting and super-resolution models.

- The PromptAware Introverted Attention appears to be a good improvement compared to the original self-attention for the text-based inpainting task.

- The experiments are sufficient, including metrics like CLIP, Acc, and PickScore. The ablation studies also indicate the effectiveness of the proposed methods.

**Weaknesses:**

- The explanation of the RASG strategy in Sec. 3.4 is intuitive but not well-supported. It would be good to provide more analysis including the scaling design choices.

- The two drawbacks mentioned in the introduction, “Prompt neglect” and “Visual context dominance,” are actually one drawback, i.e., the inpainted result is similar to the surrounding background considering the visual context but ignoring the text prompt.

- It is mentioned that “EOT is included since (in contrast with SOT) it contains information about the prompt τ” and “beneficial to normalize the scores.” It would be good to provide ablation studies to support these design choices to make this work more complete.

- As there is no user study, it would be good to provide more visual comparisons in the supplementary materials.

- The paper should discuss how unnatural the inpainted region may look sometimes. Additionally, it should discuss its limitations.

**Questions:**

I would like to see more analysis on RASG, as well as discussions on unnatural inpainting results and limitations.

---

> ### Author Response · Authors · 2023-11-19
> **We thank the reviewer for the detailed review!**
>
> We appreciate the valuable suggestions, and value the recognition of our contributions to be effective in mitigating text-alignment issues compared to prior works. Below we make some clarifications regarding weaknesses and address the suggestions.
>
> **A1** We thank the reviewer for the valuable suggestion. We have added a new section in our Appendix dedicated to RASG (Appendix E), where we include more discussion (Appendix E.2) on the effect of RASG demonstrating its advantage over the vanilla post-hoc guidance strategy with different scaling parameters. Thus, Fig. 7 shows that while classical guidance either generates unnatural results (due to out-of-distribution latents) or requires additional scaling parameter tuning, the RASG strategy of seamlessly integrating the scaled gradient into the general form of DDIM sampling is able to generate prompt-aligned and domain-preserving results.
>
> Further (Appendix E.3) we additionally provide more discussion on our choice of RASG’s rescaling with standard deviation.
>
> **A2** After some further discussion prompted by the reviewer’s comment, we have decided to change the naming we have used for “Prompt neglect” and “Visual context dominance”.
>
> In the revision both problems are now referred to as “prompt neglect”, which is further classified based on the user’s perspective of neglect, namely “background dominance” (​​when nothing is generated, just the background is coherently filled in) and “nearby object dominance” (when known objects are continued to the inpainted region while ignoring the prompt).
>
> **A3** We apologize for the confusion: the design choices coming from the both statements “EOT is included since (in contrast with SOT) it contains information about the prompt $\tau$” and “beneficial to normalize the scores” are not claimed to be essential in our PAIntA contribution. Their choice was purely motivated from logical judgements:
>
> 1. We include EOT and exclude SOT since accordion to the CLIP’s ViT based text encoder architecture uses masked attention resulting in SOT token embedding be independent from the prompt (hence does not contain prompt info), while EOT token embedding is the aggregation of all the previous embeddings, hence contains the information about the prompt $\tau$. As we want to calculate the similarities of pixels with the prompt it is logical to consider the whole information about this prompt so we choose to include EOT’s index in $ind(\tau)$. However after several experiments with EOT excluded from $ind(\tau)$ we did not observe significant differences so we kept our initial setting.
> 2. After obtaining the similarities $c_j$ normalization is required to get scaling factors from [0,1]. However the scaling can be done in different ways: min-max scaling, median-scaling with clipping, etc. Here we also tested min-max scaling which produced reasonable outputs comparable to our chosen median-scaling with clipping, hence we kept our initial choice of median scaling with clipping.
>
> **A4** Motivated by this suggestion of the reviewer, we have added **both**: a user study (Appendix C) and more visual comparisons (Fig 14, Appendix) in the revised version of our supplementary material. For details/settings of user study we kindly ask the reviewer to take a look at our supplementary material.
>
> **A5** Following this valuable suggestion, we added the Limitations section at the end of Appendix, where we discuss the possibility of unnatural and illogical results and why such cases can happen. As discussed in the added Limitations section and shown in Fig 12, unnatural results can be either attributed to the base text-guided inpainting model itself (see Fig 12.a), or to the positive prompts used during visual result generation (see Fig 12.b). Specifically, in our visual result generations we use several standard positive prompts which are known to the community to produce higher quality results. Please note that the positive prompts are being used for all methods in comparison. The reason some results are unnatural is because we didn’t pick positive prompts for each image separately, and in fact we used prompts like “trending on artstation” and “masterpiece”, which shifted the results towards a more artistic domain. That’s the reason our results sometimes look unnatural. Choosing only neutral prompts wouldn’t cause issues from the side of our method (see Fig 12.b). Though, unnatural results could still be generated because of the base model in use.
>
>
> *Concerning the question by the reviewer about more analysis on RASG unnatural results and limitations:* We added a discussion section on RASG in our supplementary material (Appendix E) including the analysis of our guidance objective function, the strategy of integrating the objective function gradient into the general form of DDIM diffusion process, and std normalization. Also we added a section Limitation in the supplementary material (Appendix I), discussing also some failure cases of unnatural results.

---

### Official Review · Reviewer_BeCN · 2023-10-28

**Soundness:** 2 fair
**Presentation:** 3 good
**Contribution:** 2 fair
**Rating:** 3
**Confidence:** 5

**Summary:**

This paper aims at achieving prompt-faithful and high-resolution image inpainting. To better align with the text prompt, the authors replace self-attention layers with the proposed Prompt-Aware Introverted Attention layer and Reweighting Attention Score Guidance during sampling. Besides, the authors propose an inpainting-specific conditional super-resolution technique for high-resolution image inpainting. The authors show both quantitative and qualitative evaluations on MSCOCO and show some good results.

**Strengths:**

1. This paper is well-organized and easy to follow. The presentation is clear.

2. The authors show some techniques to improve text alignment without training for image inpainting and show some good results.

**Weaknesses:**

1\. The motivation is not convincing to me.  The authors claim that existing technique limitation comes from the visual context dominance over the prompt in self-attention. We suggest the authors visualize or make statistics to verify such a claim.

2\. The trick used for super-resolution (blending unmasked image) are not new to the community. Some existing works already leverage such technique for image inpainting [https://github.com/huggingface/diffusers/blob/main/src/diffusers/pipelines/stable_diffusion/pipeline_stable_diffusion_inpaint.py](https://github.com/huggingface/diffusers/blob/main/src/diffusers/pipelines/stable_diffusion/pipeline_stable_diffusion_inpaint.py).

3\. The authors should compare with SmartBrush and Imagen editor, which are also proposed to solve the text alignment problem.

4\. What’s the difference between the RASG and classifier guidance or self-guidance [1]?

5\. Some important details are missing to understand PAIntA and RASG fully. For example, what does $i$ mean in Eq(5)? Does it mean text token or image token? What’s the value of c_j in the final setting? What does the update operation in Figure 3(a) mean? We suggest the authors explain it by expressions. What does X0=\epsilon(I) mean in section 3.5? Is it a decoded latent image for the input image or an upscaled latent?

6\. In section 3.3 and Figure 3, the authors borrow query and key project layers from the next cross-attention module, and calculate the similarity with selected prompt tokens but still use the value from the image feature, which confuses me. Specifically, the authors consider the prompt as a query, they should also use them as value.

7\. The experiment results are not that convincing to me. Specifically,  in Figure 13 in the supplemental results, the authors show some failure cases by using only PAIntA and RASG. It is unclear which is more powerful and why they will fail. Besides, it is also not clear why combining them works better. It seems that combining PAIntA and RASG will also fail in some cases.

\### [1] Epstein D, Jabri A, Poole B, Efros AA, Holynski A. Diffusion self-guidance for controllable image generation.

**Questions:**

My major concerns come from the novelty of the proposal and the experiment results. Specifically, the authors should discuss and compare the proposed method with relevant techniques I mentioned in the Weaknesses. Besides, the authors show some failure cases caused by only using  PAIntA and RASG. It is unclear why and when they will work. Please find more details in the Weaknesses.

---

> ### Author Response · Authors · 2023-11-19
> **We thank the reviewer for the detailed feedback! (1/3)**
>
> We appreciate the valuable feedback of the reviewer and are delighted that our work was recognized as well-organized and easy to follow with training-free text-alignment improvements demonstrating good results. Below we address the weaknesses mentioned by the reviewer and we hope our clarifications will position the contributions of this paper in their whole picture. Due to space limitations we address the questions in three comments.
>
> **A1** Indeed, we hypothesize that the issue of visual context dominance is due to non-prompt awareness of self-attention operation, where the pixels in the unknown region might be influenced from the known region pixels regardless of their compatibility with the prompt. To support this we added a discussion on our PAIntA block as a separate section in Appendix where we visualize (see Fig. 6) the attention scores in the case of vanilla self-attention and PAIntA.
> As can be noticed, in the case of vanilla self-attention the unknown region latent pixels on average have high similarities with known pixels regardless of their prompt-(non)alignment and **lead to ignoring the prompt**. Whereas in the case of PAIntA we observe high similarities of the generated pixels only with themselves and with the prompt-aligned known pixels (such as in the dog example we have high attention scores between the generated pixels and the existing dog pixels, since the prompt was “dog”). As can be seen from Fig. 6, PAIntA’s updated attention scores **lead to prompt-faithful generations**.
>
> **A2**  Our idea bears similarity to blending the latents during the backward diffusion process, which is revealed to the community by such works as (Sohl-Dickstein et al.), (Avrahami et al.), however our technology contributes to the community in two aspects:
> 1. Our approach blends the **original image high-frequencies** with the **denoised prediction** $X^{pred}_0$ whereas Avrahami et al. (the code link shared by the reviewer) blend the noisy latents $X_t$ with the noisy latents of the forward diffusion process. Hence our blending uses the noise-free **original image details** during **all diffusion time-steps** which appears extremely useful for the super-resolution generation (please see high-resolution results in our Appendix).
> 2. To the best of our knowledge we are the first to introduce a technology which uses the whole high-frequency information of the known region to **upscale** the inpainted region. We believe that our designed super-resolution method will be of a great help for the inpainting community to inpaint the missing regions in high resolution images.
> Nevertheless, to make our contribution claims more clear we revised and lowered the tone of this contribution in comparison with PAIntA and RASG. Also we added more discussion (in Section 3.5) on related blending methods as from (Sohl-Dickstein et al.), (Avrahami et al.)
>
> **A3**  We understand the reviewer’s concern and would be glad to compare our approach with the mentioned works, however there are a couple of obstacles to do so. First, the mentioned approaches are not open-source, moreover, Imagen Editor is based on a private Imagen model trained on their private data, so is not reproducible. Then we wrote emails to the authors of the works and asked them to support us in performing the quantitative and qualitative comparison with their models, and we either did not get responses or we did but they could not help us with our request.
>
> Given the above-mentioned circumstances we hope the reviewer can understand why this requirement is impossible to meet.
>
> Nevertheless we would like to also highlight that our approach is **completely training-free** whereas SmartBrush and Imagen Editor heavily benefit from large-scale training data. Our advantage in this aspect becomes especially notable when new text-guided inpainting models appear in the community with higher quality generations (such as DreamShaper 8 inpainting https://civitai.com/models/4384?modelVersionId=131004). Then, to benefit from them, our method can simply apply its plug-and-play components on top of them, whereas training-based approaches will require at least fine-tuning.

---

> ### Author Response · Authors · 2023-11-19
> **We thank the reviewer for the detailed feedback! (2/3)**
>
> **A4** Classifier guidance (Dhariwal & Nichol (2021)) and self-guidance (Epstein et al. (2023)), as vanilla post-hoc sampling strategies may lead the latents to “become out-of-distribution” (as noted by Chefer et al. (2023)) due to their additional gradient term in the diffusion sampling process. As a result some generations may become unnatural and out of the domain. To this end RASG proposes an effective way of seamless integration of the guidance objective gradient into the general form of DDIM which is shown to prevent from the distribution shift mentioned above. To support the effectiveness of RASG we compare its sampling strategy with the vanilla classifier-guidance strategy (with different guidance scales) using the same guidance objective function. Fig. 7, Appendix clearly shows the advantage of RASG.
>
> Although we emphasize RASG’s in-domain sampling strategy contribution as a difference with classifier guidance or self-guidance, however RASG also proposes its own objective function for guidance, designed especially for inpainting leveraging the open-vocabulary segmentation property of the cross-attention layers to not only guide the generations to be prompt-aligned but also correspond to the shape. We also provide a discussion on our objective function design choice in Appendix of this revision (see Appendix E.1 and Fig. 10).
>
> Additionally, we add a brief overview on existing post-hoc guidance methods in our Related Work section to enable the reader to better understand the RASG contributions. Also we would like to emphasize that we performed a deep analysis on RASG and present in Appendix E of the revised supplementary material.
>
>
> **A5** We apologize for being unclear in some points of the paper, and we address the mentioned ones here. We also kindly ask the reviewer to look at the revised version of our paper for the full picture.
> 1. In Eq. (5) we define a new matrix $\tilde{A}\_{self}$, so $i$ and $j$ are just the row and column indices respectively. As the shape of the matrix is $hw\times hw$ (and we are considering the self-attention block) both $i$ and $j$ indicate image tokens, and the number $(\tilde{A}\_{self})\_{ij}$ will measure the $c_j$**-rescaled** similarity between two image tokens positioned in the locations $i$ and $j$.
> 2. The values $c_j$ are not hyperparameters but are defined according to the similarity of the **image token** $j$ and the **prompt** $\tau$. The definition is in the next paragraph after Eq. (5): $c_j = \sum_{k\in ind(\tau)} (S_{cross})_{jk}$ followed by the median-normalization.
> 3. The update operation in Fig. 3 (a) just means obtaining the new matrix $\tilde{A}_{self}$ by Eq. (5), we apologize for a redundant block in Fig. 3 (a) and already updated it in the revised version.
> 4. Concerning $X_0 = \mathcal{E}(I)$ in Sec. 3.5: here $I$ is the original image, and $\mathcal{E}$ is the **encoder** of VQ-GAN used in Stable Diffusion as an autoencoder (also shown in Fig. 2). In other words, $X_0 = \mathcal{E}(I)$ is the **encoded latent image** in the stage of **Stable Diffusion super-resolution**.
>
>
> **A6**  PAIntA is designed to replace the self-attention mechanism, so its **only goal** is to get **better** self-attention matrix $\tilde{A}\_{self}$ in order to multiply it (after SoftMax) with the values from the image feature (as for self-attention). However to perform the modification $A_{self}\mapsto \tilde{A}\_{self}$ we use each $j^{th}$ **image feature token and the prompt** $\tau$ **similarity**, which we denote by $c_j$. For computing these (image token, whole prompt) similarities we use the next cross-attention projection layer **weights** (since they are already trained and can be used to find the similarities between feature tokens and prompt tokens). As a result 1) we calculate the similarity of feature tokens with selected prompt tokens (indexed by $ind(\tau)$) to get the scaling factor $c_j$ of the columns of $A_{self}$, 2) we use the new updated self-attention matrix $\tilde{A}\_{self}$ and old self-attention values to get the output of PAIntA.

---

> ### Author Response · Authors · 2023-11-19
> **We thank the reviewer for the detailed feedback! (3/3)**
>
> **A7** We thank the reviewer for this comment, and would like to highlight that we presented more discussions on both PAIntA and RASG in our supplementary material (Sections D and E respectively) describing the unique purposes of PAIntA and RASG. So
> 1. PAIntA enhances the self-attention scores to **gain more prompt faithful results**. However the attention score enhancement is designed to only prevent the background or nearby object dominance over the prompt when generating the missing region. Sometimes just eliminating the undesired influence of the known region is not enough for prompt-aligned generations, hence we also leverage a post-hoc guidance mechanism such as RASG.
> 2. RASG applies a post-hoc guidance to improve the text alignment even further, however **to avoid undesired latent distribution shifts** it introduces the strategy of combining the guidance with the general form of DDIM process. Using only RASG may also lead to non prompt-aligned results since the undesired influence of the known region (through self-attention layers) may be so strong that even directly guiding the generation via RASG will fail to generate the prompt objects.
> Therefore we conclude that the contributions of PAIntA and RASG are complementary to each other and best used in combination (we kindly refer to Fig. 22 for visual grounding). We also added more discussion and explanation on the ablation study of PAIntA and RASG in the supplementary material (see Appendix F).

---

> > ### Comment · Reviewer_BeCN · 2023-11-22
> > **Thanks for the authors' response**
> >
> > Thank you for the detailed response, including the additional illustrations and clarifications. Although some of my concerns have been addressed, there are still some remaining points:
> >
> > Q4. The authors claim that the key difference of the RASG strategy compared to classifier guidance sampling and self-guidance sampling is the shift in the predicted x_0 instead of the estimated score. However, to validate the effectiveness of this new design, the authors should provide convincing proof and conduct extensive ablation studies. Additionally, the mention of the open-vocabulary segmentation property of the cross-attention modules lacks references or an introduction. It would be beneficial to provide more context to clarify this claim.
> >
> > Q7. I remain skeptical about the results of PAIntA-only and RASG-only. Specifically, the existing baseline stable diffusion inpainting will also have successes or failures for the same mask and prompt, suggesting that PAIntA-only and RASG-only may also encounter failures in certain cases. I am uncertain about the possibility of cherry-picking favorable cases in the combined approach, which could lead to an unfair comparison. It is still unclear when PAIntA-only and RASG-only will fail and the extent of improvement and success rate that can be achieved.
> >
> > Considering the aforementioned concerns, I have decided to maintain my initial decision.